# A patatin-like phospholipase mediates *Rickettsia parkeri* escape from host membranes

Gina M. Borgo [1,2], Thomas P. Burke [2,3], Cuong J. Tran[1,2], Nicholas T. N. Lo [1], Patrik Engström [2,4] & Matthew D. Welch [2✉]

*Rickettsia* species of the spotted fever group are arthropod-borne obligate intracellular bacteria that can cause mild to severe human disease. These bacteria invade host cells, replicate in the cell cytosol, and spread from cell to cell. To access the host cytosol and avoid immune detection, they escape membrane-bound vacuoles by expressing factors that disrupt host membranes. Here, we show that a patatin-like phospholipase A2 enzyme (Pat1) facilitates *Rickettsia parkeri* infection by promoting escape from host membranes and cell-cell spread. Pat1 is important for infection in a mouse model and, at the cellular level, is crucial for efficiently escaping from single and double membrane-bound vacuoles into the host cytosol, and for avoiding host galectins that mark damaged membranes. Pat1 is also important for avoiding host polyubiquitin, preventing recruitment of autophagy receptor p62, and promoting actin-based motility and cell-cell spread.

[1] Division of Infectious Disease and Vaccinology, School of Public Health, University of California, Berkeley, Berkeley, CA, USA. [2] Department of Molecular and Cell Biology, University of California, Berkeley, Berkeley, CA, USA. [3] Present address: Department of Microbiology & Molecular Genetics, School of Medicine, University of California, Irvine, Irvine, CA, USA. [4] Present address: Primordial Genetics, San Diego, CA, USA. ✉email: welch@berkeley.edu

Spotted fever group (SFG) *Rickettsia* species are Gram-negative, obligate intracellular bacteria that infect tick vectors and can be transmitted to vertebrate hosts[1]. SFG *Rickettsia* that can cause disease in humans include *R. rickettsii*, the causative agent of Rocky Mountain spotted fever, a disease characterized by high fever, neurological symptoms, organ failure, and possible fatality if left untreated[2,3]. Disease-causing SFG *Rickettsia* also includes species such as *R. parkeri*, which causes milder eschar-associated rickettsiosis characterized by fever and a skin lesion (eschar) at the site of the tick bite, with no documented fatalities[2,4,5]. Because *R. parkeri* can be studied under biosafety level 2 (BSL2) conditions, it is emerging as a model for understanding the molecular determinants of SFG *Rickettsia* pathogenicity.

*R. parkeri* targets macrophages[4–8] as well as endothelial cells[6,7,9] during infection in humans and animal models. Upon invasion of host cells, bacteria escape from the primary vacuole into the cytosol, where they replicate[10,11]. Bacteria then initiate actin-based motility and move to the plasma membrane, where they enter into protrusions that are engulfed by neighboring cells[12]. This necessitates another escape event from a double-membrane secondary vacuole into the cytosol, completing the intracellular life cycle[10,11]. Other bacteria with a similar life cycle utilize pore-forming proteins and phospholipases to escape from the primary and/or secondary vacuole. For example, *Shigella flexneri* uses the IpaB-IpaC translocon to form pores that facilitate membrane rupture[13–18]. *Listeria monocytogenes* utilizes the cholesterol-dependent cytolysin listeriolysin O (LLO)[19–22] and two phospholipase C enzymes, PlcA and PlcB, to escape from primary and secondary vacuoles[19,23–27]. It is likely that *Rickettsia* also utilizes at least one protein that can directly disrupt the vacuolar membrane to mediate escape.

*Rickettsia* genomes encode two types of phospholipase enzymes, phospholipase D (PLD) and up to two patatin-like phospholipase $A_2$ (PLA$_2$) enzymes (Pat1 and Pat2)[28,29]. Genes encoding PLD and Pat1 are present in all sequenced *Rickettsia* species, whereas the gene encoding Pat2 is absent from the genome of *R. parkeri* and most other SFG species. PLD is dispensable for vacuolar escape, as a *pld* mutant in *R. prowazekii* showed no delay in escape[30], even though exogenous PLD expression in *Salmonella enterica* was sufficient to facilitate escape[31]. In contrast, evidence suggests a possible role for PLA$_2$ enzymes in escape. For example, PLA$_2$ activity from *R. prowazekii* targeted host phospholipids throughout infection[32,33]. Furthermore, pre-treatment of bacteria with either a PLA$_2$ inhibitor, or antibodies that recognize Pat1 or Pat2 or other PLA$_2$ enzymes, reduced plaque number for both *R. rickettsii*[34–36] and *R. typhi*[28,37], and increased colocalization of *R. typhi* with the lysosomal marker LAMP-1[37]. This suggests that Pat1 and Pat2 are important for infection and avoidance of trafficking to the lysosome. Nevertheless, the role of PLA$_2$ enzymes in rickettsial vacuolar escape has remained unclear.

Phospholipase activity and escape from the vacuole may also be important to enable downstream life cycle events, such as actin-based motility, which requires access to actin in the host cell cytosol. Another is avoidance of anti-bacterial autophagy (also called xenophagy). Autophagy can be initiated via polyubiquitination of cytosolic bacteria[38–40] and subsequent recruitment of autophagy receptors[41] such as p62 (also known as Sequestome 1 (SQSTM1))[42–44] and NDP52 (nuclear dot protein 52; also known as calcium-binding and coiled-coil domain 2 (CALCOCO2))[42,45,46]. Autophagy receptors recognize polyubiquitinated bacteria and interact with microtubule-associated protein 1A/1B-light chain 3 (LC3), which marks nascent and mature autophagosomal membranes that enclose bacteria and deliver them to the lysosome[38,47,48]. Bacterial phospholipases may facilitate autophagy avoidance through manipulation of phospholipids needed for autophagosome formation, such as with *L. monocytogenes* PlcA targeting of phosphatidylinositol 3-phosphate (PI(3)P) to block LC3 lipidation[49,50].

Autophagy can also be initiated by membrane damage to the bacteria-containing vacuole, which exposes glycans internalized from the host cell surface that are recognized by host cytosolic galectin (Gal) proteins[51]. Gal3 and Gal8 can target damaged vacuolar compartments during infection with *L. monocytogenes*[52,53] and *S. flexneri*[52,54,55], as well as with bacteria that reside in membrane-bound compartments such as *Legionella pneumophila*[56], *S. enterica*[52,55], *Coxiella burnetti*[57], and *Mycobacterium tuberculosis*[58]. Membrane remnants marked by Gal3 or Gal8 colocalize with polyubiquitin[54,58,59], autophagy receptors p62[54,58] and NDP52[55,57], and LC3[54,55,57,58]. Nevertheless, it remains unknown if rickettsial phospholipases are important to evade autophagy.

In this work, we characterize a *R. parkeri* mutant with a transposon insertion in the single predicted PLA$_2$-encoding gene *pat1*. We find that Pat1 is critical throughout infection for escaping host membranes, avoiding targeting by autophagy, and spreading to neighboring cells. These results suggest that Pat1 is a key bacterial factor involved in interacting with host membranes and avoiding detection in host cells.

## Results

**Pat1 possesses PLA$_2$ activity**. Patatin-like PLA$_2$ enzymes implicated in rickettsial infection have been shown to require a cofactor present in host cell lysates for enzymatic activity[28,37]. *R. parkeri* Pat1 has a patatin-like phospholipase domain with conserved amino acid residues required for catalytic activity[37] (Fig. S1a), suggesting it is also a PLA$_2$ enzyme. To determine if *R. parkeri* Pat1 has PLA$_2$ activity, we purified recombinant maltose-binding protein (MBP) tagged Pat1 (MBP-Pat1), as well as Pat1 with the catalytic serine residue at position 50 mutated to alanine (MBP-Pat1S50A), and MBP alone as a control. Purified proteins were incubated with a fluorogenic PLA$_2$ substrate (Red/Green BODIPY PC-A2) to measure enzymatic activity. MBP-Pat1 alone exhibited PLA$_2$ activity, whereas MBP-Pat1S50A and MBP exhibited little to no detectable activity (Fig. 1a). We also tested the effect of adding lysate from Vero African green monkey kidney epithelial cells on PLA$_2$ activity. The addition of Vero cell lysates enhanced the activity of MBP-Pat1, but had little to no effect on MBP-Pat1S50A or MBP (Fig. 1a). These data demonstrate that *R. parkeri* Pat1 possesses PLA$_2$ activity that is enhanced by a host factor or factors.

**Pat1 is important for infection of host cells and contributes to virulence in mice**. Next, to determine the role of Pat1 during infection, we used an *R. parkeri* mutant with a transposon insertion in the *pat1* gene (*pat1*::Tn) that was previously isolated in a screen for mutants with reduced plaque size[60]. We complemented the *pat1*::Tn mutation by generating a strain (*pat1*::Tn *pat1*⁺) that contains a second transposon encoding full-length *pat1* plus the intergenic regions immediately 5′ and 3′ to the gene (predicted to contain the native promoter and terminator) (Fig. 1b). Using an antibody that recognizes *R. parkeri* Pat1 by western blotting, we observed a band at the predicted molecular weight of 55 kD for Pat1 in wild-type (WT) bacteria, no corresponding band in the *pat1*::Tn mutant, and a restoration of the band in the *pat1*::Tn *pat1*⁺-complemented mutant (Fig. 1c). The absence of detectable Pat1 protein in the *pat1*::Tn strain suggests it is a null mutant. Because the *pat1*::Tn mutant was initially identified based on its small-plaque phenotype, we next compared plaque area for WT, mutant, and complemented mutant strains. In comparison with WT, the *pat1*::Tn mutant showed

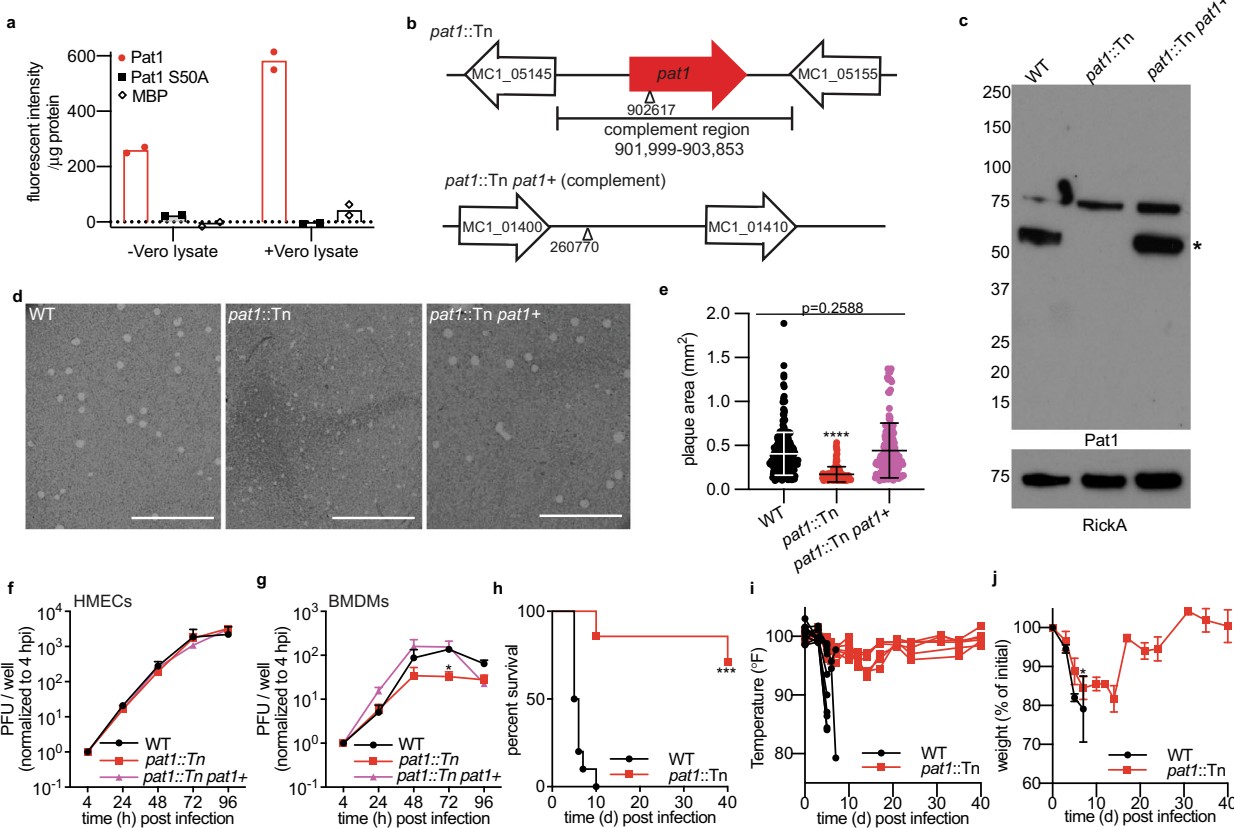

**Fig. 1 Pat1 is a PLA₂ enzyme important for infection of cells and in mice. a** Graph of fluorescence emissions intensity of fluorogenic PLA₂ substrate Red/Green BODIPY PC-A2 at 535 nm after a 2 h incubation with the indicated protein. Bar represents mean from $n = 2$ independent experiments, each with three technical replicates. **b** Genomic loci of *pat1* (top) and *pat1* insertion site for complementation (bottom). Triangle represents transposon insertion sites, and nucleotide numbers indicate the position in the *R. parkeri* genome. Genes upstream and downstream are included to show intergenic regions. **c** Western blot of purified *R. parkeri* strains, WT, *pat1*::Tn, and complemented strain (*pat1*::Tn *pat1⁺*), probed with anti-Pat1 antibody; anti-RickA was used as a loading control. Pat1 has a predicted size of 55kD (asterisk, beneath a higher molecular weight background band that is observed in all samples). The image represents results from three independent experiments. The numbers on left are molecular weight in kD. **d** Images of plaques stained with neutral red at 6 dpi (WT and *pat1*::Tn *pat1⁺*) and 8 dpi (*pat1*::Tn). Scale bar 10 mm. Represents results from two independent experiments. **e** Plaque areas in Vero cells infected with WT (7 dpi), *pat1*::Tn (8 dpi), and complemented strain (7 dpi) ($n = 2$ independent experiments; WT $n = 263$ plaques, *pat1*::Tn $n = 109$ plaques, *pat1*::Tn *pat1⁺* $n = 141$ plaques). Data are mean ± SD; ****$p < 0.0001$ relative to WT (one-way ANOVA with Dunnett's post hoc test). **f** Bacterial abundance of WT, *pat1*::Tn, and complemented strain in (**f**) HMECs ($n = 3$ independent experiments), MOI 0.01 and (**g**) BMDMs (n = 3 independent experiments), MOI 0.5. Data in (**f**, **g**) are mean ± SEM; *p*-values were significantly different in (**g**) at 72 hpi (WT vs *pat1*::Tn $p = 0.0154$, WT vs *pat1*::Tn *pat1* + $p = 0.6871$, *pat1*::Tn vs *pat1*::Tn *pat1*::Tn $p = 0.3479$) (two-way ANOVA with Tukey's post hoc test). **h** Survival of *Ifnar1⁻/⁻;Ifngr1⁻/⁻* mice infected intravenously (i.v.) with $5 \times 10^6$ WT or *pat1*::Tn mutant ($n = 10$ mice for WT, $n = 7$ mice for *pat1*::Tn, $n = 2$ independent experiments). Data were analyzed using a log-rank (Mantel-Cox) test, ***$p = 0.0001$. **i** Temperature changes over time in i.v. infection of *Ifnar1⁻/⁻;Ifngr1⁻/⁻* mice with $5 \times 10^6$ WT or *pat1*::Tn mutant bacteria; graphs represent data from individual mice. **j** Weight change over time expressed as percent change from initial weight in i.v. infection of *Ifnar1⁻/⁻;Ifngr1⁻/⁻* mice with $5 \times 10^6$ WT or *pat1*::Tn mutant bacteria. Data are mean ± SEM and were analyzed using a two-way ANOVA from 0 to 7 dpi, *$p = 0.0308$. Source data are provided as a Source Data file. .

significantly smaller plaques, and plaque area was rescued in the complemented mutant (Fig. 1d, e). This demonstrates that the observed reduction in plaque area was caused by loss of *pat1*.

To further determine if Pat1 plays a role in bacterial replication, growth curves measuring plaque-forming units (PFU) were performed in a human microvascular endothelial cell line (HMECs) and primary murine bone marrow-derived macrophages (BMDMs). There were no differences in bacterial replication kinetics for WT, *pat1*::Tn, and *pat1*::Tn *pat1⁺*-complemented strains in HMECs (Fig. 1f). In BMDMs, bacterial replication was impaired for the *pat1*::Tn mutant at 72 hpi (Fig. 1g). These data indicate that Pat1 is important to sustain normal bacterial growth rates in BMDMs but not HMECs.

We next examined the contribution of Pat1 to virulence in vivo using mice lacking the receptors for both IFN-I (*Ifnar1*) or IFN-γ (*Ifngr1*) (*Ifnar1⁻/⁻;Ifngr1⁻/⁻* double knock out mice), which succumb to infection with WT *R. parkeri* and can be used to investigate the importance of bacterial genes to virulence[8,61]. Mice infected intravenously (i.v.) with $5 \times 10^6$ PFU WT bacteria showed a rapid drop in temperature and body weight following infection and did not survive past day 8 (Fig. 1h–j). In contrast, mice infected i.v. with the *pat1*::Tn mutant maintained a steady temperature following infection, showed an initial drop in weight that stabilized around 2 weeks post infection before increasing, and the majority survived until the end of the experiment (day 40) (Fig. 1h–j). These results indicate that Pat1 is an important virulence factor.

**Pat1 promotes efficient escape from the vacuole post-invasion.**
Because *R. typhi* Pat1 and Pat2 had previously been implicated in
avoidance of trafficking to lysosomes[37], we sought to determine if
the *R. parkeri pat1*::Tn mutant was impaired in its ability to
escape from the primary vacuole during infection. To evaluate the
role of Pat1 in vacuolar escape, we used transmission electron
microscopy (TEM) to investigate whether host membranes sur-
rounded intracellular bacteria at 1 hpi in HMECs. This time point
was chosen because prior studies reported escape from the
vacuole by 30 min post infection (mpi) for *R. typhi*[37], *R.
prowazekii*[30], and *R conorii*[62]. At 1 hpi, significantly more WT
bacteria were found free in the cytosol (74%) compared with the
*pat1*::Tn mutant (38%) (Fig. 2a, b). Moreover, significantly fewer
WT bacteria were found within membranes (25% in single
membranes, 1% in double membranes) in comparison with the
*pat1*::Tn mutant (50% in single membranes, 12% within double
membranes).

To further quantify vacuolar escape at later time points (1.5
and 4 hpi), we differentiated cytosolic from vacuolar bacteria by a
primary digitonin permeabilization of the plasma membrane to
detect cytosolic bacteria followed by a secondary saponin
permeabilization of all intracellular membranes to detect all
bacteria including those enclosed in a vacuole[14,63,64]. We found
that in HMECs, a significantly higher percentage of *pat1*::Tn
mutant bacteria were in vacuoles at 1.5 hpi (Fig. 2c, d) than WT
and *pat1*::Tn *pat1*+ (Fig. 2c, d). We also observed more vacuolar
*pat1*::Tn mutant bacteria compared to WT in BMDMs at 1.5 hpi
(Fig. 2e). In contrast, at 4 hpi in HMECs, the number of vacuolar
bacteria was similar for all three strains, suggesting that the
*pat1*::Tn eventually escapes into the cytosol. Differences in
vacuolar localization were not due to differences in the invasion
of host cells, as both WT and the *pat1*::Tn mutant entered cells
with similar kinetics when measured by staining intracellular and
extracellular bacteria (Fig. S2a, b). Collectively, these results
indicate that Pat1 is important for the rapid escape of bacteria
from the vacuole into the cytosol.

Next, we investigated the trafficking of bacteria to LAMP-1
positive compartments at 1–2 hpi, because the inability to escape
the vacuole can lead to a higher frequency of bacteria being
trafficked to late endosomes and lysosomes. We observed
increased colocalization of LAMP-1 with the *pat1*::Tn mutant at
2 hpi in HMECs (Fig. S3a, b) and at both 1 and 2 hpi in BMDMs
(S3c, d). These results suggest that, following invasion, Pat1
facilitation of escape from membranes enables avoidance of
trafficking to the lysosome.

Lastly, we hypothesized that the increased localization of the
*pat1*::Tn mutant within membranes impaired access to the
cytosol, particularly to the pool of actin, interfering with actin-
based motility. To test this hypothesis, we quantified the number
of bacteria with actin tails at 30 mpi and 1 hpi using fluorescence
microscopy. Approximately 3–4% of WT bacteria were associated
with actin tails, in keeping with previous reports[65,66]. The
frequency of *pat1*::Tn mutant association with actin tails was half
that of WT at both time points (Fig. 2f, g). These results suggest
that failure of the *pat1*::Tn mutant to escape from the vacuole
leads to a reduction in actin-based motility. To confirm that the
reduced frequency of actin-based motility resulted from bacteria
being trapped within membranes, we used hypotonic shock
(alternating treatment with hypertonic and then hypotonic
solutions) to lyse primary vacuoles[67,68] and more efficiently
deliver bacteria to the cytosol. When cells infected with WT
bacteria were subjected to hypotonic shock at 5 mpi, there was no
significant increase in the percentage of bacteria with actin tails at
30 mpi, suggesting that WT bacteria optimally access the cytosol
following invasion (Fig. 2h). In contrast, hypotonic shock
significantly increased the percentage of *pat1*::Tn mutant bacteria

with actin tails. These results confirm that the reduced frequency
of actin-based motility in the *pat1*::Tn mutant is due to
entrapment in the primary vacuole.

**Pat1 contributes to autophagy avoidance.** The presence of a
marked fraction (12%) of *pat1*::Tn mutant bacteria in double-
membrane compartments at 1 hpi could not be explained by
failure to escape from the vacuole, suggesting the possibility that
bacteria were targeted by host cell autophagy. Because an initial
step of anti-bacterial autophagy is recognition and ubiquitylation
of the bacterial surface[40], we first tested for bacterial association
with polyubiquitin in infected HMECs at 0–2 hpi. Whereas fewer
than 2% of WT bacteria were polyubiquitin-positive from 0–2
hpi, the percentage of polyubiquitin-positive *pat1*::Tn mutant
bacteria was significantly higher and increased (from about 6% at
0 hpi to about 16% at 1 hpi), before falling slightly (Fig. 3a, b).
Complementation of the *pat1*::Tn mutant reduced the percentage
of polyubiquitin-positive bacteria to levels seen with WT (Fig. 3c).
Higher percentages of polyubiquitin-positive *pat1*::Tn mutant
bacteria compared to WT were also observed in BMDMs at 1 hpi
(Fig. 3d). This suggests that Pat1 reduces the recognition of
bacteria by the host ubiquitylation machinery.

To further examine whether bacteria were targeted by the
autophagy machinery, we examined the recruitment of autophagy
receptors p62 and NDP52, as well as LC3 at 1 hpi, the time point
with the most polyubiquitin-positive bacteria. Compared with
WT (fewer than 2% stained with these markers), markedly more
of the *pat1*::Tn mutant were positive for p62 (10%) and NDP52
(6%) (Fig. 4a–c). Moreover, more of the *pat1*::Tn mutant bacteria
colocalized with LC3 at 1 and 2 hpi (Fig. 4d, e). As noted above,
the increased recruitment of LC3 to the *pat1*::Tn mutant in
HMECs preceded increased colocalization of the mutant with
LAMP-1, a marker for late endosomal and lysosomal compart-
ments (Fig. S3a, b). These results suggest that Pat1 is important
for counteracting the recruitment of autophagy receptors and
targeting to autophagosomes and lysosomes.

Because previous studies suggested that Pat1 is secreted into
the host cell[37], we also sought to further ascertain whether Pat1
(or other factors produced by WT bacteria) was counteracting
ubiquitylation and targeting by the autophagy machinery by
acting locally on the bacterium producing the protein, and/or by
acting at a distance on other bacteria. To test this, we co-infected
HMECs with WT bacteria expressing 2xTagBFP and with
*pat1*::Tn mutant bacteria, and quantified colocalization of
*pat1*::Tn bacteria with polyubiquitin, NDP52, and p62. The
*pat1*::Tn mutant exhibited significantly reduced colocalization
with polyubiquitin and p62 (but not NDP52) in co-infected cells
compared with cells infected with the *pat1*::Tn mutant only
(Fig. 4f). These results indicate that the presence of WT bacteria
counteracts targeting of the *pat1*::Tn mutant bacteria by
ubiquitylation and recruitment of the autophagy machinery.

**Pat1 antagonizes bacterial association with damaged mem-
branes that recruit Gal3, Gal8 and NDP52.** It remained unclear
whether polyubiquitin and the autophagy machinery were asso-
ciated with bacteria enclosed in damaged vacuolar membranes or
those free in the cytosol. To determine whether polyubiquitin,
NDP52, and p62 were present at damaged vacuoles at 1 hpi, we
quantified the percentage of bacteria staining for polyubiquitin,
NDP52, or p62, in cells transiently expressing Gal3-mCherry or
Gal8-mCherry to mark damaged membranes[52,55] (Fig. 5a, b). A
small fraction (0.5%) of WT bacteria colocalized with Gal3-
mCherry or Gal8-mCherry, whereas significantly more *pat1*::tn
mutant bacteria colocalized with Gal3-mCherry or Gal8-mCherry
(~2.5%) (Fig. 5c, d). A significantly higher fraction of the *pat1*::Tn

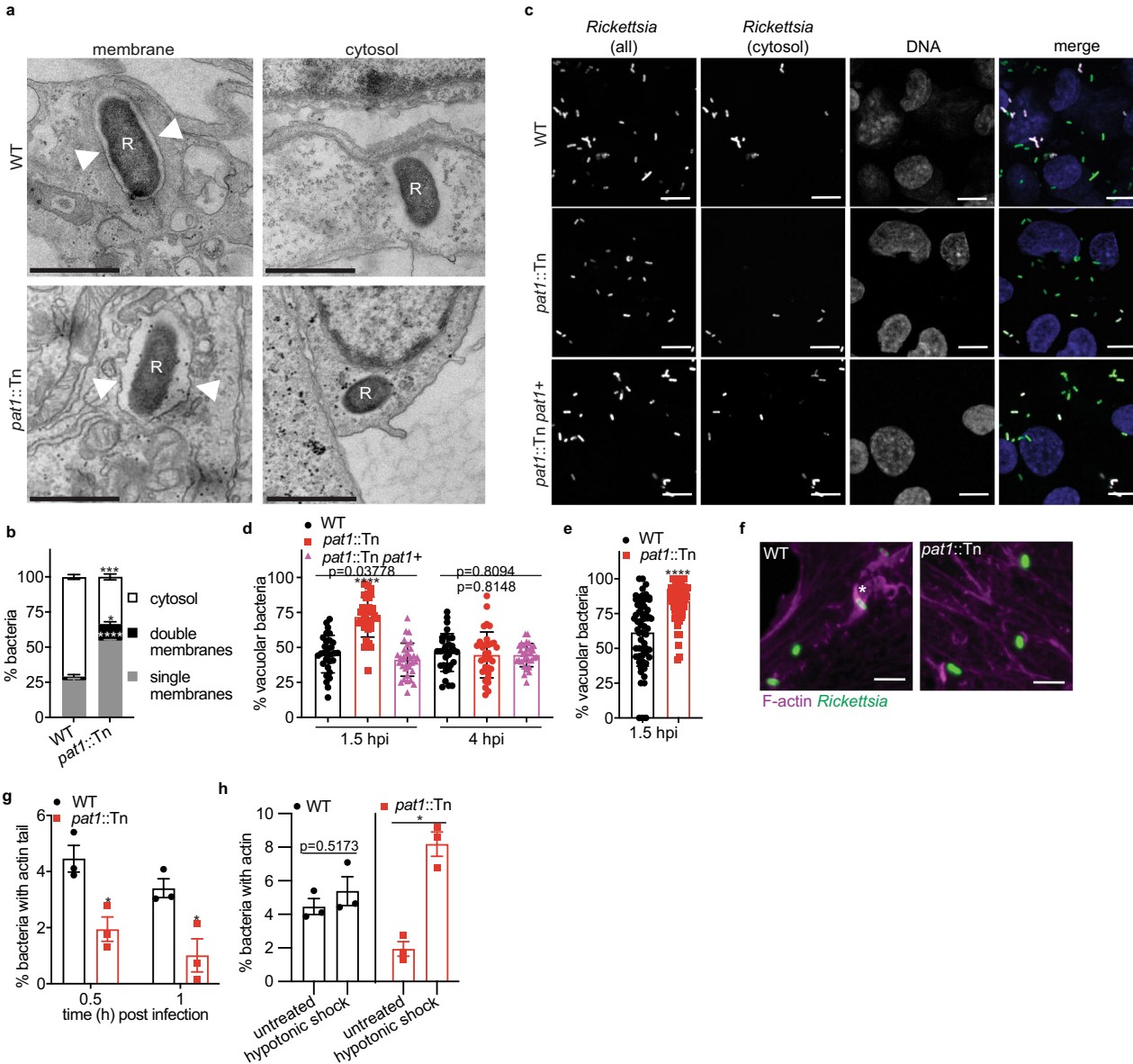

**Fig. 2 Pat1 facilitates escape from single and double membrane compartments following invasion. a** TEM images of WT and *pat1*::Tn mutant bacteria in HMECs at 1 hpi. "R" indicates *R. parkeri* and arrowheads point to membrane surrounding the bacteria. Scale bar 1 µm. **b** quantification of (**a**), percentage of single and double membrane-bound or cytosolic bacteria ($n = 3$ independent experiments, >50 bacteria quantified), MOI 5. Data are mean ± SEM ($n = 3$ independent experiments); ****$p < 0.0001$ (single membranes) ***$p = 0.0002$ (cytosol) *$p = 0.0115$ (double membranes) relative to WT (unpaired *t*-test (two-tailed). **c** Images of cytosolic bacteria (magenta) in HMECs infected with WT, *pat1*::Tn, or *pat1*::Tn *pat1*+ bacteria (green in merge) at 1.5 hpi. Scale bar 5 µm. **d** Quantification of (**c**), percentage of vacuolar bacteria at 1.5 and 4 hpi in HMECs, MOI 5 ($n = 2$ independent experiments; WT $n = 32$ fields (1.5 hpi), 33 fields (4 hpi), *pat1*::Tn $n = 33$ fields (1.5 hpi, 4 hpi), *pat1*::Tn *pat1*$^+$ $n = 33$ fields (1.5 hpi), 32 fields (4 hpi). **e** Quantification of vacuolar bacteria at 1.5 hpi in BMDMs, MOI 5 ($n = 3$ independent experiments; WT $n = 65$ fields, *pat1*::Tn $n = 77$ fields). Data in (**d, e**) represent individual fields imaged representing $n > 1500$ bacteria for each strain and timepoint; bars are mean ± SD. Data in (**d**) ****$p < 0.0001$ relative to WT (one-way ANOVA with Dunnett's post hoc test. Data in (**e**) ****$p < 0.0001$ relative to WT (unpaired *t*-test (two-tailed)). **f** Images of early (30 mpi) actin tails in HMECs, (magenta, F-actin; green, *R. parkeri*), asterisk indicates bacterium with actin tail. Scale Bar 3 µm. **g** Quantification of (**f**), percentage of bacteria with actin tails at 30 mpi and 1 hpi (>500 bacteria counted per time point), MOI 5. Data are mean ± SEM ($n = 3$ independent experiments); *$p = 0.01738$ (0.5 hpi), $p = 0.02461$ (1 hpi) relative to WT (unpaired *t*-test (two-tailed)). **h** Percentage of bacteria with actin in untreated cells or cells that have undergone hypotonic shock treatment to lyse vacuoles (images not shown; >1000 bacteria counted per strain/condition), MOI 5. Data are mean ± SEM ($n = 3$ independent experiments); *$p = 0.0325$ relative to untreated (paired *t*-test (two-tailed)). Source data are provided as a Source Data file.

mutant bacteria that stained for NDP52 also colocalized with either Gal protein (~50%) (Fig. 5e). Some *pat1*::Tn mutant bacteria that stained for p62 also colocalized with Gal3-mCherry (~5%) or Gal8-mCherry (~10%) (Fig. 5e). We rarely observed colocalization of the *pat1*::Tn mutant bacteria with both poly-ubiquitin and either Gal protein (Fig. 5e). Interestingly, although

the *pat1*::Tn mutant was more frequently associated with Gal3-mCherry or Gal8-mCherry, in *pat1*::Tn mutant-infected cells we observed fewer clusters of Gal3-mCherry (Fig. S4a, c) or Gal8-mCherry (Fig. S4b, d) not associated with bacteria, consistent with reduced overall membrane damage compared with cells infected with WT bacteria.

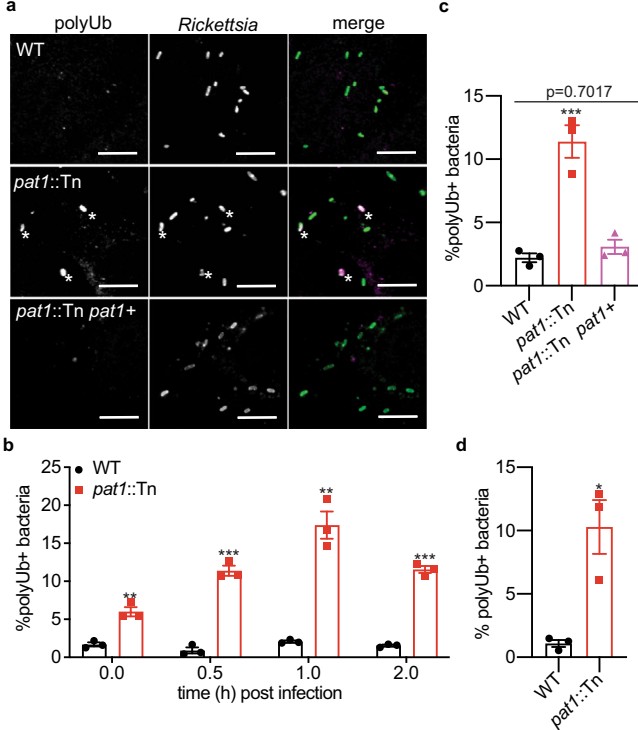

**Fig. 3 Pat1 contributes to avoidance of polyubiquitin recruitment.**
**a** Images of polyubiquitin (polyUb; magenta in merge) in HMECs infected with WT, pat1::Tn, and pat1::Tn pat1 + bacteria (green in merge) at 1 hpi. Asterisk denotes colocalization between bacterium and polyUb. Scale bar 5 μm. **b** Quantification of (**a**), percentage of polyUb-positive bacteria at the indicated time points (>1000 bacteria counted per strain/time point), MOI 3. Data are mean ± SEM; ***p = 0.00016 (0.5 hpi), p = 0.00003 (2 hpi), **p = 0.00281 (0 hpi), p = 0.00105 (1 hpi) relative to WT (unpaired t-test, two-tailed). **c** Percentage of polyUb-positive bacteria in HMECs infected with WT, pat1::Tn, or complemented mutant (pat1::Tn pat1+) at 1 hpi, MOI 3 (>1000 bacteria counted per strain). Data are mean ± SEM; ***p = 0.0005 relative to WT (one-way ANOVA with Dunnett's post hoc test). **d** Percentage of polyUb-positive bacteria at 1 hpi in BMDMs (>1000 bacteria counted per strain), MOI 5. Data are mean ± SEM; *p = 0.0126 relative to WT (unpaired t-test, two-tailed). All data represent n = 3 independent experiments. Source data are provided as a Source Data file.

To test whether NDP52 is associated with bacteria or damaged membranes, we performed the hypotonic shock treatment (Fig. 2e) to release bacteria from host vacuoles. Cells infected with WT and pat1::Tn bacteria were subjected to hypotonic shock treatment at 5 mpi, and then at 30 mpi, we quantified the number of bacteria that colocalized with NDP52. Hypotonic shock significantly reduced the percent colocalization of the pat1::Tn with NDP52 (from ~6% in untreated cells to ~1% in treated cells), while fewer than 1% of WT bacteria colocalized with NDP52 regardless of treatment (Fig. 5f). Together, these results support the conclusion that Pat1 promotes efficient escape from vacuolar membranes and enables avoidance of targeting by Gal3, Gal8, and NDP52.

**Pat1 facilitates actin-based motility and spread into neighboring cells late in infection.** Pat1 is important for plaque formation, suggesting that it may function in cell–cell spread. To initially assess if Pat1 is important for spread, we used an infectious focus assay, in which the number of infected host cells per focus of infection was quantified at 28 hpi to measure spread efficiency[69]. Compared with WT bacteria (~4.5 cells per focus), the pat1::Tn

mutant infected significantly fewer cells (~3.5 cells per focus) (Fig. 6a, b). This suggested that Pat1 is important for spread. To further assess cell–cell spread, we carried out a "mixed cell" assay (67) in which "primary" A549 cells stably expressing the TagRFP-T-farnesyl plasma-membrane marker (A549-TRTF) were infected for 1 h, detached from the plate, and mixed with unlabeled "secondary" A549 cells (Fig. 6c, d). The percentage of bacteria in the primary A549-TRTF cell and secondary cell were quantified at 32 hpi. We observed that 50% of WT bacteria were found in primary cells and 50% had spread into secondary cells. In contrast, ~85% of pat1::Tn mutant bacteria remained in primary cells and only ~15% had spread to secondary cells (Fig. 6c, e). This confirms that Pat1 is important for cell–cell spread.

Because our data indicated that Pat1 facilitates cell–cell spread, we further investigated whether Pat1 influences actin-based motility, which is known to contribute to spread[65,66]. We found that the pat1::Tn mutant formed significantly fewer actin tails compared to WT bacteria at 24 hpi and 48 hpi (Fig. 6f, g), suggesting fewer bacteria-initiated actin-based motility. Complementation of the pat1::Tn (pat1::Tn pat1+) mutant restored the frequency of actin tail formation to WT levels (Fig. S5a, b). In the mixed cell assay, which distinguishes between primary and secondary cells, a higher percentage of WT bacteria in both the primary and secondary cell assembled actin (mostly as actin tails but also as "clouds" of actin surrounding the bacteria) compared with pat1::Tn mutant bacteria (Fig. S5c). The observed differences between WT and the pat1::Tn mutant were not due to differences in the localization of the R. parkeri protein Sca2, which is important for actin-based motility and cell–cell spread[66,70] (Fig. S5d). Taken together, these results suggest that Pat1 is important for the frequency of bacterial actin-based motility, and hence bacterial spread to neighboring cells.

**Pat1 is important for avoiding double membranes during cell–cell spread.** Because Pat1 played an important role in escaping the primary vacuole following the invasion, we hypothesized that it also played a role in escaping the secondary vacuole following cell–cell spread. To test this, we imaged infected HMECs by TEM at 48 hpi and quantified the percentage of intracellular bacteria free in the cytosol or within membranes. Significantly more pat1::Tn mutant bacteria were surrounded by double membranes (~60%) in comparison with WT bacteria (~25%) (Fig. 7a, b). The double membranes we observed were often discontinuous, with the mutant remaining mostly enclosed and WT bacteria having very few surrounding membrane fragments. This suggests that Pat1 plays a role in escaping from membranes later in infection.

We next sought to further distinguish whether bacteria surrounded by double membranes were in secondary vacuoles that result from cell–cell spread, or other double-membrane structures involved in autophagy. To determine targeting by autophagy at 48 hpi, we used fluorescence microscopy to assess whether bacteria colocalized with polyubiquitin, p62 or NDP52 at 48 hpi. Significantly more of the pat1::Tn mutant colocalized with p62 and NDP52 than WT, although the overall percentages were low in all cases (Fig. 7c). Moreover, the percentage of bacteria that colocalized with these markers was lower than at 1 hpi (compare with Figs. 3b, 4c). Interestingly, polyubiquitin labeling was not significantly different between WT and pat1::Tn mutant bacteria, suggesting the observed differences in colocalization with autophagy receptors p62 and NDP52 between WT and the pat1::Tn mutant were not mediated by differences in polyubiquitin recruitment. These data suggest that a minor fraction of double membranes observed by TEM at 48 hpi are autophagosomes and Pat1 plays a minor role in autophagy avoidance at late time points.

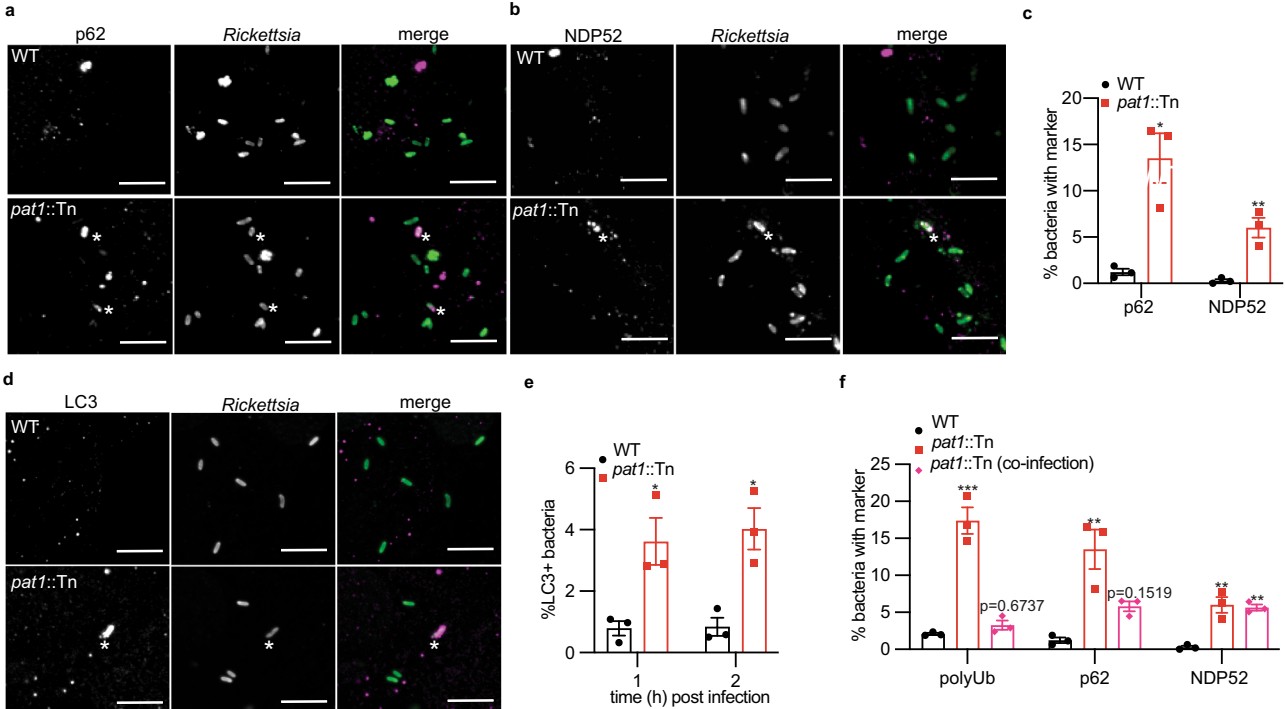

**Fig. 4 Pat1 enables evasion of recognition by autophagy.** Images of autophagy receptors **a** NDP52 (left; magenta in merge) and **b** p62 (right; magenta in merge) in WT and *pat1*::Tn (green in merge) infected HMECs. Asterisk denotes colocalization between bacterium and p62 or NDP52. Scale bars in (**a**, **b**) 5 μm. **c** Quantification of (**a**) and (**b**), percentage of bacteria staining for NDP52 or p62 at 1 hpi (>1000 bacteria counted per strain for each marker), MOI 3. Data are mean ± SEM; **$p = 0.0060$, *$p = 0.0354$ relative to WT (unpaired *t*-test (two-tailed)). **d** Images of LC3 (magenta) in HMECs infected with WT and *pat1*::Tn bacteria (green) at 1 hpi. Asterisk denotes colocalization between bacterium and LC3. Scale bars 5 μm. **e** Quantification of percent bacteria staining for LC3 at 1 hpi (images in (**d**)) and 2 hpi (images not shown), (>1500 bacteria counted per strain for each time point), MOI 3. Data are mean ± SEM; *$p = 0.0244$ (1 hpi), $p = 0.0124$ (2 hpi) relative to WT (unpaired *t*-test (two-tailed)). **f** Percentage colocalization of bacteria with polyUb, NDP52, and p62 in HMECs infected with WT, *pat1*::Tn mutant, or co-infected with WT and *pat1*::Tn mutant (>700 bacteria counted for each strain/ marker), MOI 5. For co-infections, quantification is for *pat1*::Tn bacteria only. Data are mean ± SEM; ***$p = 0.0001$, **$p = 0.0030$ (p62), $p = 0.0016$ (NDP52, *pat1*::Tn), $p = 0.0023$ (NDP52, co-infection) relative to WT (one-way ANOVA with Dunnett's post hoc test). All data represent $n = 3$ independent experiments. Source data are provided as a Source Data file. .

To determine whether there instead were differences in escape from secondary vacuole, we used the fluorescence microscopy-based mixed cell assay described above, in which infected primary cells stably expressing TagRFP-T-farnesyl were infected for 1 h and then mixed with uninfected and unlabeled secondary cells (Fig. 6c, d). Fewer than 1% of WT bacteria that spread from primary into secondary cells were colocalized with the plasma membrane marker from the primary cell (Fig. 7d), suggesting that these bacteria had escaped the secondary vacuole. In contrast, of the *pat1*::Tn mutant bacteria that spread into secondary cells, ~12% colocalized with the plasma membrane marker from the primary cell. These results suggest that a significant fraction of double-membrane structures seen in the TEM images are secondary vacuoles and confirm that Pat1 is important for escaping from these vacuoles.

## Discussion

The ability of *Rickettsia* species to escape the vacuole and avoid host membranes is a critical facet of their life cycle. Here, we demonstrate that the *R. parkeri* patatin-like phospholipase, Pat1, is important for virulence in a mouse model of infection. At the cellular level, we find that Pat1 enables bacterial escape from host membranes throughout infection. Pat1 mediates efficient exit from primary vacuoles following the invasion, helping *R. parkeri* avoid detection by host galectins and autophagy adapter NDP52. Pat1 further enables cytosolic bacteria to avoid recruitment of polyubiquitin and autophagy adapter p62. As the infection

progresses, Pat1 facilitates spread into neighboring cells and escape from the secondary vacuole. Altogether, these data suggest Pat1 plays a primary role in vacuolar escape and is of importance at multiple steps of the *Rickettsia* life cycle that involve manipulating host membranes.

Our work shows that Pat1 is important for virulence upon i.v. infection in a mouse model that succumbs to *R. parkeri* infection[8,61]. This is consistent with other bacterial factors involved in vacuolar escape being important virulence factors in animal models of disease. For example, *L. monocytogenes* mutants lacking LLO are avirulent upon i.v. infection in a mouse model and single *L. monocytogenes* PLC mutants are also diminished in virulence[27]. Our data suggest that Pat1 plays an important role in *R. parkeri* pathogenesis.

Using TEM and confocal microscopy, we found that Pat1 mediates escape from both single and double-membrane compartments in host cells. At early time points, *pat1*::Tn mutant bacteria were more frequently surrounded by single membranes following the invasion, likely to be primary vacuoles derived from the host cell plasma membrane. Consistent with a failure to fully escape the primary vacuole, the *pat1*::Tn mutant also showed a reduced exposure to the cytosol (as measured by digitonin permeabilization), reduced frequency of actin-based motility, and increased trafficking to LAMP-1-positive compartments. We also found the *pat1*::Tn mutant had increased localization to double-membrane structures at later time points when bacteria are spreading to neighboring cells. These structures are likely to be

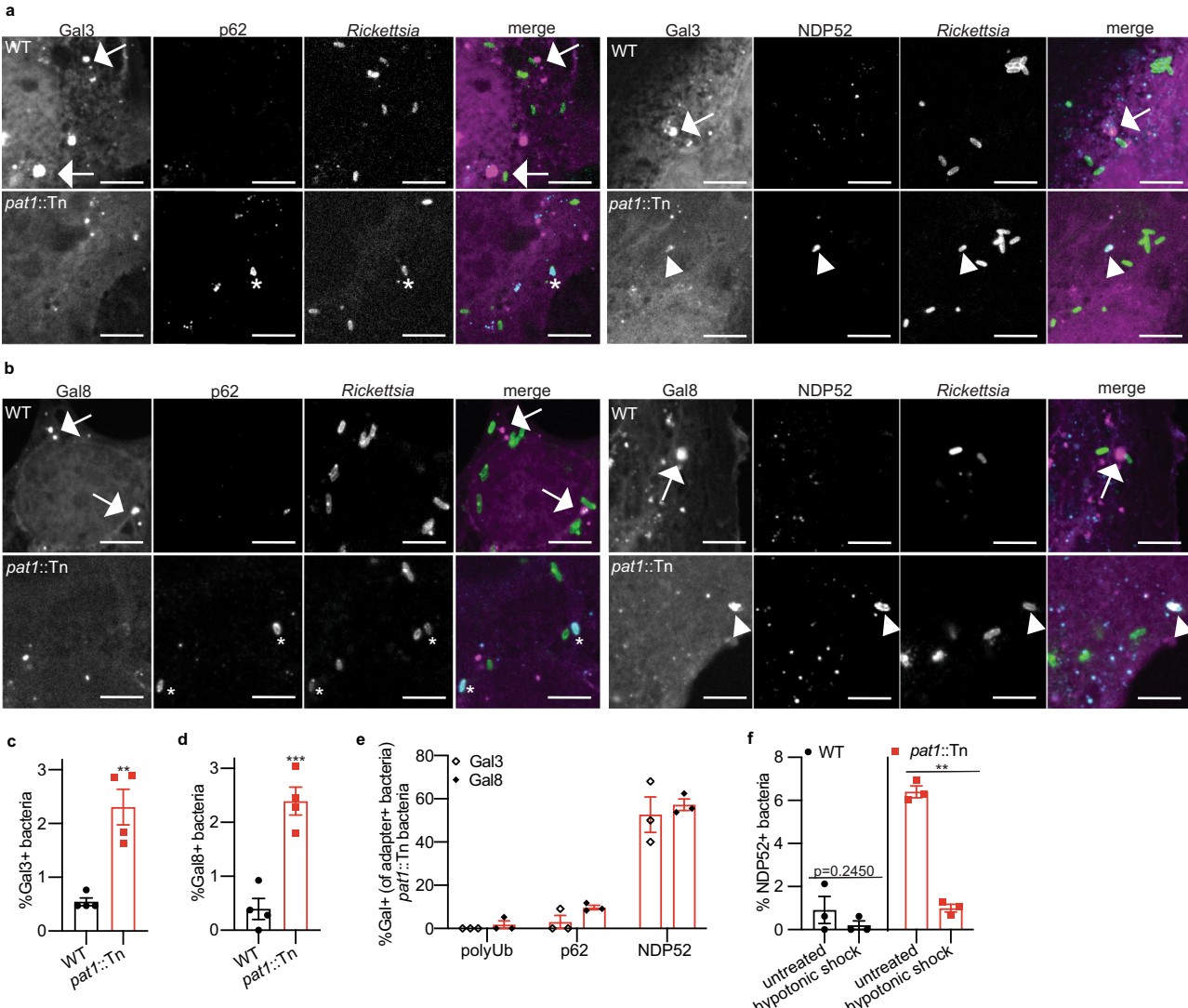

**Fig. 5 Pat1 is important for avoiding bacterial association with damaged membranes. a** Images of Gal3-mCherry (magenta in merge) and autophagy receptors p62 (left; cyan in merge) and NDP52 (right; cyan in merge) in HMECs infected with WT or *pat1*::Tn bacteria (green in merge) at 1 hpi. Arrows indicate large Gal3 positive clusters near bacteria, asterisk indicates bacteria that are p62 positive and Gal3 negative, arrowheads indicate colocalization between NDP52, Gal3, and bacteria. **b** Images of Gal8-mCherry (magenta in merge) and autophagy receptors p62 (left; cyan in merge) and NDP52 (right; cyan in merge) in HMECs infected with WT or *pat1*::Tn bacteria (green in merge) at 1 hpi. Arrows indicate large Gal8 positive clusters near bacteria, asterisk indicates bacteria that are p62 positive and Gal8 negative, arrowheads indicate colocalization between NDP52, Gal8, and bacteria. Scale bar for (**a**) 5 μm, (**b**) 3 μm. **c** Quantification of (**a**), percentage of bacteria positive for Gal3 (>500 bacteria counted per strain), MOI 5. **d** Quantification of (**b**), percentage of bacteria positive for Gal8 (>500 bacteria counted per strain), MOI 5. Data in (**c, d**) are mean ± SEM; ***$p = 0.0008$ **$p = 0.0020$ relative to WT (unpaired *t*-test (two-tailed)). **e** Quantification of (**a**) and (**b**), percent of polyUb, p62, or NDP52 positive *pat1*::Tn bacteria that are also positive for Gal3 (open diamond) or Gal8 (filled diamond)(50–60 polyUb, p62, or NDP52 positive bacteria in Gal+ cells were counted for both Gal3 and Gal8), MOI 5. **f** Percent of bacteria positive for NDP52 in untreated cells or cells that undergo hypotonic lysis of vesicles (>1000 bacteria counted per strain/condition), MOI 5. Data are mean ± SEM; **$p = 0.0063$ relative to untreated (paired *t*-test (two-tailed)). **c, d** Data represent $n = 4$ independent experiments, **e, f** data represent $n = 3$ independent experiments. Source data are provided as a Source Data file.

secondary vacuoles, as only a small portion colocalized with autophagy receptors p62 or NDP52. Pat1 was previously suggested as a candidate for escape from the vacuole due to its phospholipase activity[28,37] and the observation that *R. typhi* pretreated with PLA₂-blocking antibodies (which could block surface-associated but not secreted Pat1) caused increased colocalization with LAMP-1[37]. Our results provide genetic confirmation of this role. Several other bacterial phospholipases mediate membrane rupture[71], including *L. monocytogenes* PLCs[19,26,27,71], *Clostridium perfringens* alpha-toxin (PLC)[71–73], and *Psuedomonas aeruginosa* ExoU[71,74]. Similarly, lecithin:cholesterol acyltransferase (LCAT) enzymes from pathogenic

protists, including *Plasmodium berghei* phospholipase (PbPL)[75,76] and *Toxoplasma gondii* TgLCAT[77], have PLA₂ and acyl transferase activity that facilitate the breakdown of the parasitophorous vacuole. Phospholipases are also used by nonenveloped viruses to breech the endosome[78], including parvovirus capsid protein VP1 which has PLA₂ activity[79], and host PLA₂ group XVI which is recruited by picornaviruses to endosomes for genome translocation[80]. Thus, the role of Pat1 in vacuolar breakdown and escape is similar to the functions of phospholipases in other intracellular pathogens.

Pat1 is dispensable for growth in HMECs, whereas it is important for normal growth kinetics in BMDMs, suggesting that additional

 

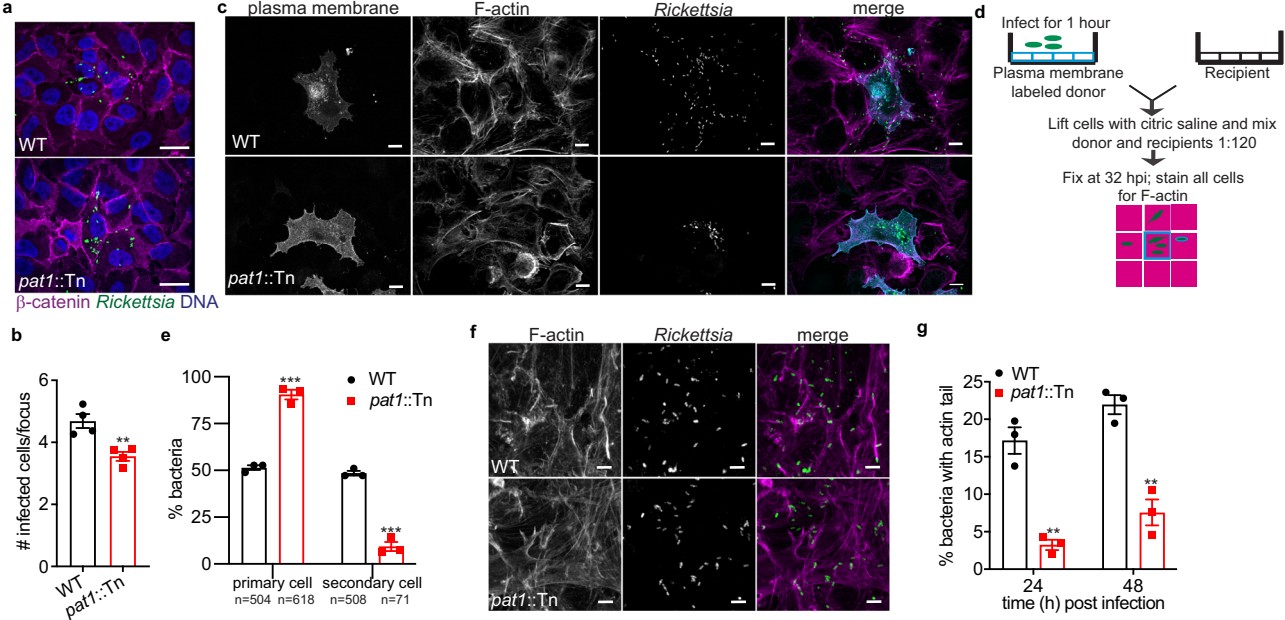

**Fig. 6 Pat1 is important for cell-cell spread and facilitates actin-based motility. a** Images of infectious foci formed by WT or *pat1*::Tn mutant in A549 cells at 28 hpi (magenta, β-catenin; green, bacteria; blue, nuclei). Scale bar 10 μm. **b** Quantification of (**a**), number of infected cells per focus (30–45 foci quantified total), MOI 0.05. Data are mean ± SEM; **$p = 0.0054$ relative to WT (unpaired t-test (two-tailed)). **c** Images of mixed cell assay depicted in (**d**) showing plasma membrane (A549-TRTF; cyan in merge), F-actin (magenta in merge), and bacteria (green in merge), adapted from[67]. Scale bar 10 μm. **e** Percent bacteria in primary and secondary cells quantified from (**c**), MOI 4. The total number of bacteria counted in the primary and secondary cells for each strain is shown below the x-axis label. Data are mean ± SEM; ***$p = 0.0002$ (primary cell), $p = 0.0001$ (secondary cell) relative to WT (unpaired t-test (two-tailed)). **f** Images of actin tails (F-actin; magenta in merge) and bacteria (green in merge) in HMECs at 24 hpi. Scale bar 5 μm. Images represent results from three independent experiments. **g** Percentage of bacteria with actin tails at 24 hpi and 48 hpi in HMECs (>1000 bacteria counted per strain/time point), MOI 0.5. Data are mean ± SEM; **$p = 0.00186$ (24 hpi), $p = 0.00257$ (48 hpi) relative to WT (unpaired t-test (two-tailed)). **b** Data represent $n = 4$ independent experiments and **e**, **g** data represent $n = 3$ independent experiments. Source data are provided as a Source Data file.

host mechanisms restrict rickettsial infection in BMDMs. Cell type differences in the ability to restrict bacterial growth have been observed previously for *R. parkeri*[81]. Also, *L. monocytogenes* LLO is required for vacuolar escape in murine, but not human cell lines[19,21,25,27]. The importance of Pat1 to growth in BMDMs suggest that further characterization of the role of Pat1 in different cell types, both in cell culture and in vivo, will be critical to understanding how Pat1 promotes survival in the host.

The fact that Pat1 mutants are recoverable indicates that other *R. parkeri* proteins help rupture vacuolar membranes and gain access to nutrients in the cytosol. Consistent with this notion, the *pat1*::Tn mutant colocalizes more frequently with damaged membranes marked by Gal3 and Gal8 and eventually escapes into the cytosol at a frequency similar to WT bacteria. Pat1 must therefore share functional redundancy with other proteins that also mediate escape. This is similar to the redundancy between *L. monocytogenes* PLC enzymes, which also have overlapping roles in escape, with double mutants deficient in both PlcA and PlcB showing more severe defects in escape[19,26,27] and growth[27,49] than single mutants. Other *Rickettsia* factor(s) involved in escape may include TlyC, a putative hemolysin[31,82], which could function analogously to LLO. Pat2, a second PLP, may also have an overlapping role with Pat1 in those species with the gene that encodes this protein, such as the typhus group species *R. typhi*[28,37]. In addition to the membranolytic proteins, Risk-1, a phosphatidylinositol 3-kinase, was recently reported to manipulate early trafficking events important for invasion, vacuolar escape, and autophagy[83]. Factors such as these are likely to function with Pat1 to allow complete escape from vacuolar membranes into the cytosol.

A key role for Pat1 in promoting efficient escape from damaged host membrane remnants is to enable subsequent

avoidance of targeting of these membranes by Gal proteins and the autophagy machinery. Consistent with this, we observed that the *pat1*::Tn mutant colocalizes more frequently with Gal3 or Gal8 and the autophagy receptor NDP52. Interestingly, Gal3, but not Gal8, has been found to promote *L. monocytogenes* replication during infection by suppressing autophagy[53]. During Group A *Streptococcus* infection, Gal3 has also been shown to prevent recruitment of Gal8 and parkin, which themselves play antibacterial roles[84]. Whether differential recruitment of Gal3 and Gal8 leads to different outcomes during *Rickettsia* infection remains unknown. Nevertheless, our results suggest that Pat1 enables efficient vacuolar escape which allows for reduced detection and targeting by components of the autophagy pathway.

We found that Pat1 also plays a role in avoiding targeting by autophagy for bacteria that are not associated with damaged membranes. In support of this notion, the *pat1*::Tn mutant was subject to polyubiquitylation and p62 recruitment without co-recruitment of Gal proteins. To what extent a role for Pat1 in avoidance of autophagic targeting of bacteria independent of membrane damage is a secondary consequence of Pat1 function in vacuolar escape or a more direct function of Pat1 remains unclear. Pat1 may thus augment other rickettsial autophagy-avoidance mechanisms, which include lysine methylation of OmpB and OmpB-mediated shielding of bacterial surface from polyubiquitylation[81,85]. In this role, Pat1 might function in a similar manner to PlcA from *L. monocytogenes*, which reduces PI(3)P levels to block autophagosome formation and stall autophagy[49,50]. PlcA/B is secreted and can act at a distance, as it was previously shown that a *plcA/B* mutant can be rescued by co-infection with by WT *L. monocytogenes*[50], and we also observed

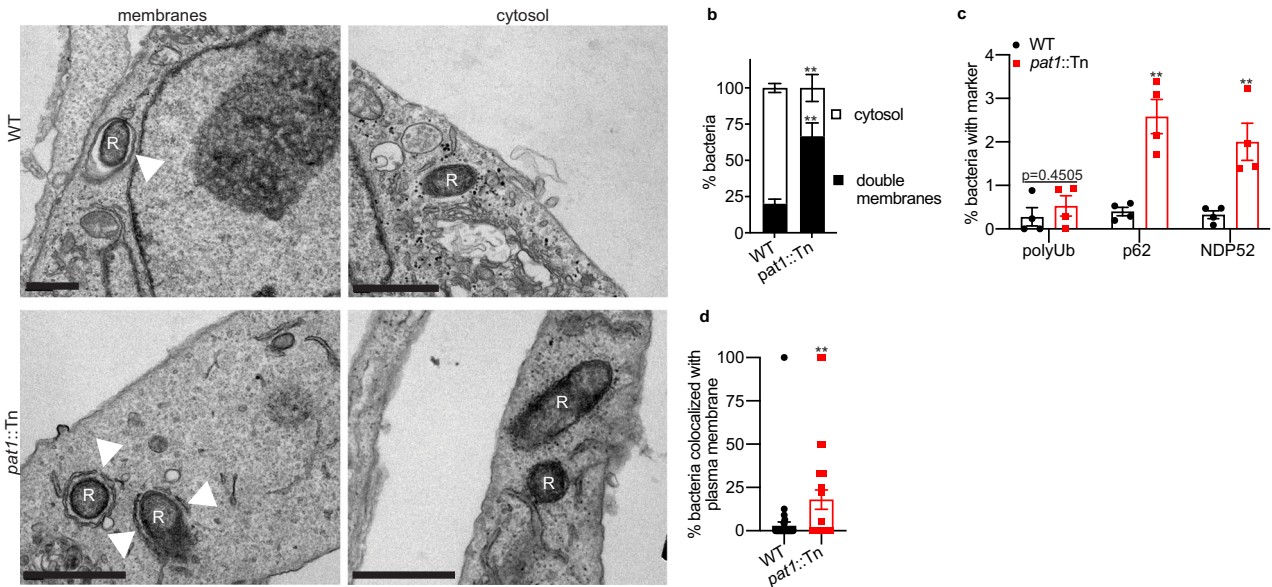

**Fig. 7 Pat1 is important for escape from the secondary vacuole. a** TEM images of WT and *pat1*::Tn mutant bacteria in HMECs at 48 hpi. "R" indicates *R. parkeri* and arrowheads point membranes surrounding the bacteria. Scale bar 1 µm. Images represent results from three independent experiments. **b** Percentage of bacteria in double-membrane compartments or in the cytosol (*n* = 3 independent experiments, >300 bacteria quantified), MOI 0.1. Data are mean ± SEM; **\*\*p* = 0.0089 relative to WT (unpaired *t*-test (two-tailed)). **c** Percentage of WT and *pat1*::Tn mutant bacteria colocalizing with polyUb, p62, and NDP52 (all *n* = 4 independent experiments) at 48 hpi in HMECs (>1000 bacteria counted per strain/marker), MOI 0.5. Quantification was done using fluorescence microscopy with the same antibodies as Figs. 3a and 4a, b. Data are mean ± SEM; **\*\*p* = 0.0017 (p62), *p* = 0.0067 (NDP52) relative to WT (unpaired *t*-test (two-tailed)). **d** Percentage of bacteria in the secondary cell that colocalize with the plasma membrane from the primary cell (measured using fluorescence microscopy) in mixed cell assays from Fig. 6c at 32 hpi (*n* = 3 independent experiments), MOI 4. Data represent individual fields imaged that contained bacteria in secondary cells (*n* = 46 fields for WT, *n* = 27 fields for *pat1*::Tn; bacterial number is in Fig. 6e). Data are mean ± SEM; **\*\*p* = 0.0026 (Mann–Whitney (two-tailed)). Source data are provided as a Source Data file.

rescue of a *R. parkeri pat1*::Tn mutant by co-infection with WT bacteria. Thus, Pat1 might also target early and/or regulatory aspects of autophagy.

We further found that Pat1 is important for cell-cell spread, including in late actin-based motility and escape from the secondary vacuole (the latter is discussed above). The *pat1*::Tn mutant formed fewer actin tails in primary infected cells and exhibited reduced spread into neighboring cells when compared with WT, consistent with the known role for motility in cell-cell spread of SFG *Rickettsia*[65,66,70]. The *pat1*::Tn mutant also formed fewer actin tails in secondary infected cells following cell-cell spread, suggesting at least some of the reduction in actin tails at late time points is due to reduced escape from the secondary vacuole. One key contribution of Pat1 to actin-based motility is to mediate escape from the vacuole, allowing recruitment of the host actin machinery to the surface of the bacteria. However, it remains possible that Pat1 targeting of phosphoinositides (PIs) also affects actin-based motility, as PIs influence the activity of actin-binding proteins[86–88]. Moreover, Pat1 targeting of PIs at the plasma membrane could contribute to protrusion dynamics during cell-cell spread. PIs are implicated in the recruitment and function of the endocytic machinery[89]. In turn, endocytic pathways have been shown to mediate protrusion engulfment for *L. monocytogenes* and *S. flexneri*[90,91]. Pat1-mediated local membrane damage might also promote spread, as *L. monocytogenes* LLO-mediated membrane damage in the protrusion has been shown to enable exploitation of efferocytosis for spread[92]. Thus, Pat1 may play multiple roles in cell-cell spread.

Our data demonstrate that Pat1 plays important roles throughout the *R. parkeri* intracellular life cycle, with a principal role in escaping from vacuoles, and additional contributions to avoiding autophagy and enabling cell-cell spread. Whether Pat1

directly performs other critical functions during infection remains to be determined. For example, secreted Pat1 could contribute to the release of bioactive lipids such as eicosanoids derived from arachidonic acid. Eicosanoid synthesis can impact immunity, inflammation, and vascular function[93–95], and thus represents an underexplored process that may be influenced by *Rickettsia* infection and disease. Membranes are critical hubs of signaling and protein-protein interactions and *R. parkeri*, like other intracellular pathogens, has likely evolved diverse ways of manipulating membranes. Further studies of Pat1 function will elucidate how $PLA_2$ enzymes facilitate microbial adaptation to host cells and could reveal previously unappreciated strategies of membrane manipulation by bacterial pathogens.

## Methods

**Mammalian cell lines.** Mammalian cell lines were obtained from the UC Berkeley Cell Culture Facility and grown at 37 °C with 5% $CO_2$. Vero cells (African green monkey kidney epithelial cells, RRID:CVCL_0059) were grown in Vero media (DMEM with high glucose (4.5 g/L; Gibco, 11965-092) and 2% FBS (GemCell, 100500) for culturing or 5% FBS for plaque assays (described below)). A549 cells (human lung epithelial cells, RRID:CVCL_0023) were grown in A549 media (DMEM (Gibco, 11965-092) with high glucose (4.5 g/L) and 10% FBS (ATLAS, F-0500-A)). A549 cells stability expressing a farnesyl tagged TagRFP-T (A549-TRTF) to mark the plasma membrane was described previously[69] and were also maintained in A549 media. HMEC-1 cells (human microvascular endothelial cells RRID:CVCL_0307) were grown in HMEC media (MCDB 131 media (Sigma, M8537) supplemented with 10% FBS (HyClone, SH30396.03), 10 mM L-glutamine (Sigma, M8537), 10 ng/ml epidermal growth factor (Corning, 354001), 1 µg/mL hydrocortisone (Spectrum Chemical, CO137), and 1.18 mg/mL sodium bicarbonate).

Primary murine bone marrow-derived macrophages (BMDMs) were generated from the bone marrow of C57BL/6 mice as described previously[61,81]. Cells were thawed the day before infection and grown in media containing DMEM with high glucose (4.5 g/l; Gibco, 11965-092), 20% FBS (HyClone, SH30088.03), 10% colony-stimulating factor (obtained from supernatant from NIH 3T3 fibroblasts), 1%

sodium pyruvate (Gibco, 11360-070), and 0.1% β-mercaptoethanol (Gibco, 21985-023).

**Plasmid construction**. For complementation of *pat1*::Tn *pat1*+ mutant, we constructed a pMW1650-Spec-*pat1* complementation plasmid. The plasmid pMW1650-Spec was derived from pMW1650[96] by deleting the gene encoding GFP and replacing the gene conferring resistance to rifampicin with the *aadA* gene from *E. coli* to confer resistance to spectinomycin[97]. Nucleotides 901,999-903,853 from *R. parkeri* genomic DNA were then amplified by PCR and inserted into pMW1650-Spec at the single PstI site. Primers used to amplify this region (5′-ATTGCGA-CACGTACTCTGCAGATCTCATACCATCATAGTTATAATATTAGC-3′ and 5′-AGAGGATCCCCATGGCTGCAGACACAGGTGTCGTCATTGTGA-3′) contained 15-bp overhang with homology to the plasmid for InFusion (Takara Bio, 638947) cloning and retained the Pst1 site. The amplified sequence contained a predicted promoter 5′ to the *pat1* coding sequence (determined using SoftBerry, BPROM prediction of bacterial promoters; http://www.softberry.com/berry.phtml?topic=bprom&group=programs&subgr oup=gfindb)[98] and several predicted transcriptional terminators 3′ to the *pat1* coding sequence (determined using WebGeSTer DB; http://pallab.serc.iisc.ernet.in/gester/)[99].

To express Pat1 in *E. coli* for antibody generation and PLA₂ activity assays, the DNA sequence encoding full-length *pat1* was amplified from *R. parkeri* genomic DNA by PCR and subcloned into a pET1 vector containing an N-terminal 6x His-tag, maltose-binding protein (MBP) tag, and TEV cleavage site (Addgene plasmid #29656). Primers used for amplifying the insert (5′-TACTTCCAATCCAATGTAGATATAAACAACAATAAGATTAGC-3′ and 5′-TTATCCACTTCCAATGAGATAACCTTGTACATCATCTGTATGC-3′) contained 15-bp overhang with homology to the plasmid for InFusion cloning (Takara Bio, 638947). The resulting plasmid was pET-M1-6xHis-MBP-TEV-Pat1. To express a mutated version of Pat1 in which the catalytic serine at position 50 is substituted to an alanine (Pat1S50A) for PLA₂ activity assays, PCR site-directed mutagenesis was performed on the *pat1* gene using the following overlapping primers for the S50A mutation: 5′-ATATTCGATTTTACTGGAGGGACCGCTGTTGGAGGACTTATTTC-TATTTTG-3′ and 5′-CTCACAAGTAGGCTTACCGGTTATTT-3′. PCR products were gel purified, digested with DpnI to remove template DNA, and transformed into *E. coli* strain XL1-Blue (UC Berkeley QB3 Macrolab). Plasmid DNA was extracted from the resulting colonies and verified to contain the desired mutation by DNA sequencing. The above-mentioned plasmids were transformed into *E. coli* strain BL21 codon plus RIL-Cam^r (DE3) (UC Berkeley QB3 Macrolab).

To express Pat1 in *E. coli* for antibody affinity purification from rabbit sera, the DNA sequence encoding full-length *pat1* was amplified from pET-M1-6xHis-MBP-TEV-Pat1 by PCR and subcloned into the pSMT3 plasmid containing a 6xHis-tag upstream of the SUMO tag[100]. Primers used for amplifying insert (5′-CACAGAGAACAGATTGGTGGATCCATGGTAGATATAAACAACAATAA-GATTAG-3′ and 5′-GTGGTGGTGGTGGTGTAACTCGAGGAGATAACCTTGTACATCATCTG-TAT GC-3′) contained 15-bp overhang with homology to the plasmid for InFusion cloning (Takara Bio, 638947). The resulting plasmid, pSMT3-6x-His-SUMO-Pat1, was transformed into *E. coli* strain BL21 codon plus RIL-Cam^r (DE3) (UC Berkeley QB3 Macrolab).

To make pmCherry-N1-Gal8, full-length Gal8 was amplified from PB-CAG-mRuby3-Gal8-P2A-Zeo (Addgene plasmid #150815) and subcloned into pmCherry-N1 (Clontech, 632523). Primers used for amplifying insert were 5′-ACCGCGGGCCCGGGATCCGCCACCATGATGTTGTCCTTAAACAACC-3′ and 5′-GCGACCGGTGGATCCCCCCAGCTCCTTACTTCCAGT-3′. The forward primer contained a Kozak sequence and both primers contained 15-bp overhang with homology to the plasmid for InFusion cloning (Takara Bio, 638947). To make pmCherry-N1-Gal3, Gal3 cDNA was amplified using primers 5′-CCGGAATTCGCCACCATGGCAGACAATTTTTCGCTC-3′ and 5′-CGCGGATCCCGTATCATGGTATATGAAGCACTG-3′ and subcloned into pmCherry-N1.

**R. parkeri strains and bacterial isolation**. *R. parkeri* Portsmouth strain (WT) was provided by Dr. Christopher Paddock (Centers for Disease Control and Prevention). The *pat1*::Tn mutant was generated from this strain as described previously[60]. *R. parkeri* strains were purified by infecting confluent Vero cells in T175 flasks at an MOI of 0.05. Flasks were monitored for plaque formation and harvested when 70–80% of the cells in the flask were rounded, typically 5–7 d after infection. Cells were scraped and pelleted at 12,000 × g for 30 min at 4 °C. The pelleted cells were resuspended in ice-cold K36 buffer (0.05 M KH₂PO₄, 0.05 M K₂HPO₄, pH 7.0, 100 mM KCl, 15 mM NaCl) and transferred to a Dounce homogenizer. Repeated douncing of 60–80 strokes released intracellular bacteria and the lysed cells and bacteria were centrifuged at 200 × g for 5 min at 4 °C. The supernatant containing the bacteria was overlaid on a 30% MD-76R solution (Bracco Diagnostics, NDC 0270-0860-30) and centrifuged at 58,300 × g for 30 min at 4 °C in a SW-28 rotor to further separate host cell components from bacteria. Bacterial pellets were resuspended in brain heart infusion (BHI) media (BD Difco, 237500) and stored at −80 °C. Purified bacteria were thawed on ice from frozen stocks for infections. Purified bacteria are referred to as "30% preparations" below.

For complementation of the *pat1*::Tn mutant, small-scale electroporations were performed with pMW1650-Spec as previously described for pMW1650[60]. Following electroporation, 200 µl of bacteria per well were added to confluent Vero cells in a 6-well plate. The plate was rocked at 37 °C for 30 min in a humidified chamber. An overlay of Vero media with 5% FBS and 0.5% agarose was added to each well and incubated at 33 °C, 5% CO₂ for 24 h. To select for transformants, an overlay of Vero media with 5% FBS, 0.5% agarose, and 40 µM spectinomycin was added to the cells for plaque isolation. Individual plaques were picked and resuspended in 200 µl BHI media. To grow plaque-isolated bacteria, plaques were added to Vero cells in a T25 flask and rocked at 37 °C for 30 min. Spectinomycin was added to 40 µM final concentration, and the flasks were placed at 33 °C and monitored for bacterial growth and harvested when 70–80% of the cells in the flask were rounded, typically 5–7 d after infection. Infected cells were scraped from the flask and pelleted at 2000 × g for 5 min at room temperature followed by resuspension in K36 buffer. Cells were mechanically disrupted by vortexing at ~2900 rpm (Vortex Genie 2) with 1 mm glass beads for two 30 s pulses with 30 s incubations on ice after each pulse. Following the disruption, host cell debris was pelleted by centrifuging at 200 × g for 5 min at 4 °C. The supernatant was transferred to pre-chilled microcentrifuge tubes and spun at 10,000 × g for 2 min at 4 °C to pellet *R. parkeri*. Bacterial pellets were washed three times with cold 250 mM sucrose, then resuspended in 200 µl BHI (50 µl was frozen at −80 °C). To further expand the bacterial population, 150 µl of bead-prepped bacteria was mixed with 350 µl of Vero media and added to Vero cells in a T75 flask, rocked at 37 °C for 30 min, supplemented with Vero media to a final volume of 12 ml, and incubated at 33 °C in 5% CO₂. After 5–7 d, when 70–80% of the cells in the flask were rounded, the process of bead disruption and bacteria isolation was repeated, except the bacteria were resuspended in BHI without any sucrose washes to generate frozen stocks that were stored at −80 °C. Bacterial strains were screened by PCR for the following: (1) the presence of the original transposon using primers for the rifampicin resistance cassette (primers 5′-ATGGTAAAAGATTGGATTCCTATTTCTC-3′ and 5′-CCTTAATCTTCAATAACATGT-3′); (2) the presence of second transposon using primers for spectinomycin resistance cassette (primers 5′-TGATTTGCTGGTTACGGTGAC-3′ and 5′-CGCTATGTTCTCTTGCTTTTG-3′); and (3) the presence of *pat1*::Tn and WT *pat1* using primers amplifying *pat1* (primers 5′-GTAGATATAAACAACAATAAGATTAGC-3′ and 5′-GAGATAACCTTGTACATCATCTGTATGC-3′). Strains were also screened by assessing plaque size and Pat1 expression by western blot (procedure described below). The *pat1*::Tn *pat1*+ strain that contained the original transposon, the second transposon, plaque size similar to WT, and restored Pat1 protein levels was propagated to purify bacterial 30% preparations as described above.

The insertion site of the transposon was mapped as previously described[96]. Briefly, *R. parkeri* genomic DNA was purified from frozen 30% preparations of bacteria (~10⁸ PFU). Bacteria were thawed and centrifuged at 16,000 × g for 5 min to pellet bacteria, then genomic DNA was purified using DNeasy Blood and Tissue kit (Qiagen, 69504), according to the manufacturer's protocol for Gram-negative bacteria (except that the proteinase K incubation was done overnight). One microgram of *R. parkeri* genomic DNA was digested with HindIII (New England Biolabs, R0104T). The reaction was heat-inactivated and DNA fragments were self-ligated using T4 DNA ligase (New England Biolabs, M0202T). *E. coli* were transformed with the ligation reaction and plated onto Luria-Bertani (LB) agar plates with 40 µM spectinomycin to select for plasmids containing the pMW1650-Spec transposon and flanking regions of genomic DNA. Plasmid DNA was sequenced (primers 5′-ATCTCGCTTTACCTTGGATTCC-3′ and 5′-CTATACGAAGTTGGGCATAC-3′) to determine the genomic location of the insertion site. The insertion site was then confirmed by PCR from genomic DNA using primers that flank the insertion region (5′-AAAGCGGGAATCCAGTAAATC-3′ and 5′-GGCACAGCAGAAATTACTCTTG-3′).

**Plaque assays and growth curves**. To determine the titer of purified bacteria, 200 µl of bacteria diluted $10^{-3}$–$10^{-8}$ in Vero media were added to Vero cells grown in six-well plates. Plates were rocked at 37 °C for 30 min then overlaid with 3 ml of Vero media with 5% FBS and 0.7% agarose. For imaging plaques, neutral red (0.01% final concentration; Sigma, N6264) in Vero media with 2% FBS and 0.5% agarose was overlaid onto cells 5–7 d post infection. Because of differences in timing of plaque formation for the WT and *pat1*::Tn mutant strains, plaque counts for WT and complemented *pat1*::Tn plaques were done at ~5 d post infection, and those for *pat1*::Tn plaques were done at ~7 d post infection. Plaques were counted and imaged 24 h after the addition of neutral red.

Growth curves were carried out following infection of HMECs at an MOI of 0.01 or BMDMs at an MOI of 0.5 in 24-well plates. At each time point, media was aspirated from individual wells, cells were washed twice with sterile deionized water, 1 ml of sterile deionized water was added, and cells were lysed by repeated pipetting. Three serial dilutions of the supernatant from lysed cells in Vero media, totaling 1 ml each, were added in duplicate to confluent Vero cells in 12-well plates. Plates were spun at 300 × g for 5 min at room temperature and incubated at 33 °C overnight. The next day, media was aspirated and 2 ml of Vero media with 5% FBS and 0.7% agarose was overlaid in each well. Once plaques were visible, an overlay with neutral red was done as described above.

**Protein expression and antibody generation**. For expression of MBP-Pat1 for antibody generation or PLA$_2$ activity assays, *E. coli* strain BL21 codon plus RIL-Cam$^r$ (DE3) with plasmid pET-M1-6xHis-MBP-TEV-Pat1 was grown in LB with 25 mM glucose an OD$_{600}$ of 0.5 and expression was induced with 1 mM IPTG for 1 h at 37 °C. Bacteria were pelleted by spinning at 2034 × *g* for 30 min at 4 °C, and the pellet was resuspended in MBP lysis buffer (50 mM Tris-HCl, pH 8.0, 300 mM NaCl, 1 mM EDTA) supplemented with 1 µg/ml each leupeptin (MilliporeSigma, L2884), pepstatin (MilliporeSigma, P5318), and chymostatin (MilliporeSigma, E16), and 1 mM phenylmethylsulfonyl fluoride (PMSF, MilliporeSigma, 52332). Bacteria were flash-frozen in liquid nitrogen and stored at −80 °C. Bacterial cultures were thawed quickly and kept on ice or at 4 °C for the remaining steps. Lysozyme (Sigma, L4919) was added to a final concentration of 1 mg/ml followed by a 15 min incubation on ice. Bacteria were subjected to 8 cycles of sonication at 30% power for 12 s bursts, followed by rest on ice for 30 s. Lysed bacteria were spun at 20,000 × *g* for 30 min at 4 °C. The supernatant was passed three times over a column of 10 ml of amylose resin (New England Biolabs, E8031L). The column was washed with MBP wash buffer (50 mM Tris-HCl, pH 8.0, 300 mM NaCl) by passing 15 column volumes. Bound protein was eluted by adding 2–3 column volumes of MBP elution buffer (50 mM Tris-HCl, pH 8.0, 300 mM NaCl, 0.5 mM DTT, 10 mM maltose) to the column and collecting 500 µl fractions. Fractions were checked for eluted protein by both Bradford assay and SDS-PAGE, and fractions with the highest concentration of protein and a single band at the expected molecular weight for MBP-Pat1 or MBP-Pat1S50A were pooled and concentrated. For PLA$_2$ activity assays, purified fractions were concentrated and exchanged into PLA$_2$ reaction buffer (50 mM Tris-HCL, pH 8.9, 100 mM NaCl, 1 mM CaCl$_2$; from Invitrogen, E10217), stored in aliquots at −80 °C, and thawed once for use.

To generate rabbit anti-Pat1 antibodies, 1.7 mg of purified MBP-Pat1 was sent to Pocono Rabbit Farm and Laboratory. Immunization was carried out following their 91-day custom antibody production protocol, then extended for an additional 6 weeks for an additional boost and bleed before final exsanguination.

To affinity purify anti-Pat1 antibodies, *E. coli* strain BL21 codon plus RIL-Cam$^r$ (DE3) with plasmid pSMT3-6x-His-SUMO-Pat1 was grown, induced for protein expression, and isolated as described above. Bacterial pellets were resuspended in His lysis buffer (20 mM Tris-HCl, pH 8.0, 300 mM NaCl, 10 mM imidazole) supplemented with protease inhibitors PMSF and LPC at the same concentrations as described above. Bacteria were lysed by sonication and the lysate centrifuged as described above. The supernatant was incubated with 2.0 ml of Ni-NTA resin (Qiagen, 30210) for 1 h at 4 °C with rotation and the resin was applied to a column. The column was washed with His wash buffer (20 mM Tris-HCL, pH 8.0, 300 mM NaCl, 30 mM imidazole), and protein was eluted from the column in 500 µl aliquots with 2 column volumes of His elution buffer (50 mM NaH$_2$PO$_4$, pH 8.0, 300 mM NaCl, 250 mM imidazole). The same protocol was followed to purify 6x-His-SUMO from *E. coli* strain BL21 codon plus RIL-Cam$^r$ (DE3) transformed with the parental plasmid pSMT3. Purified 6x-His-SUMO or 6x-His-SUMO-Pat1 were coupled to NHS-activated Sepharose 4 fast flow resin (GE Healthcare, 17-0906-01) in ligand coupling buffer (200 mM NaHCO$_3$, pH 8.3, 500 mM NaCl) for 2–4 h at room temperature. To remove anti-SUMO antibodies, the 6x-His-SUMO resin was incubated with 10 ml anti-Pat1 diluted in binding buffer (20 mM Tris-HCl, pH 7.5) and incubated at 4 °C for 2 h with rotation. The flow-through was collected and added to the 6x-His-SUMO-Pat1 resin and was incubated at 4 °C for 4 h with rotation. Bound antibody was eluted with 100 mM glycine, pH 2.5, into tubes containing 120 µl 1 M Tris-HCl, pH 8.8, to neutralize to pH 7.5. Eluted fractions were dialyzed in phosphate-buffered saline (PBS; 137 mM NaCl, 2.7 mM KCl, 10 mM Na$_2$PO$_4$, 1.8 mM KH$_2$PO$_4$, pH 7.4) with 50% glycerol (pH 8.0) overnight at 4 °C, concentrated and stored at −20 °C.

**In vitro PLA$_2$ activity**. PLA$_2$ activity of MBP-Pat1, MBP-Pat1S50A, and MBP were carried out using a protocol similar to that described for activity assays with *R. typhi* Pat1 and Pat2[28,37]. The EnZ Chek Phospholipase A2 Kit (Invitrogen, E10217) was used as indicated in the manufacturer's instructions. Liposomes containing the fluorogenic substrate Red/Green BODIPY PC-A2 (1-O-(6-BODIPY 558/568-Aminohexyl)−2-BODIPY FL C5-Sn-Glycero-3-Phosphocholine) were made fresh the day of the experiment by combining 30 µl of the following in 5 ml of PLA$_2$ reaction buffer: 1 mM Red/Green BODIPY PC-A2, 10 mM dioleoylphosphatidylcholine, and 10 mM dioleoylphosphatidylglycerol. For each reaction, 50 µl of liposomes were added to 50 µl of sample containing 10 µg of each individual purified protein with or without 20 µg of Vero cell lysate (prepared as described below). The reaction was incubated at room temperature for 2 h while protected from light. Fluorescence was measured using a Tecan Infinite F200 Pro plate reader with excitation at 485 nm and emission at 535 nm. Fluorescence intensity per µg of protein was calculated by subtracting the raw fluorescence intensity of the sample from the raw fluorescence intensity of PLA$_2$ reaction buffer only and dividing by the amount of protein added. For samples incubated with Vero cell lysates, the raw fluorescence intensity of the sample was subtracted from the raw fluorescence intensity of liposomes mixed with Vero cell lysate.

Vero cell lysates were prepared as described previously[28]. Briefly, ~8 × 10$^6$ Vero cells were washed 2× with ice-cold PLA$_2$ reaction buffer. Cells were scraped into 2 ml of ice-cold PLA$_2$ reaction buffer and subjected to 4 cycles of sonication at 50% power for 5 s bursts, followed by rest on ice for 30 s. Lysed cells were centrifuged for 15 min at 4 °C at 1000 × *g* and the supernatant of Vero lysate was snap-frozen

in 200 µl and stored at −80 °C. Samples were thawed once the day of the experiment and protein concentration was determined following thaw using Bio-Rad Protein Assay Dye Reagent (Bio-Rad, 500-0006).

**Western blotting**. For detection of Pat1 in bacterial cell lysates, 30% preparation bacteria (preparation described above) were boiled in 1x SDS loading buffer (150 mM Tris pH 6.8, 6% SDS, 0.3% bromophenol blue, 30% glycerol, 15% β-mercaptoethanol) for 10 min, resolved on a 10% SDS-PAGE gel, then transferred to a PVDF membrane (Millipore, IPFL00010). The membrane was blocked overnight at 4 °C in TBS-T (20 mM Tris, 150 mM NaCl, pH 8.0, 0.1% Tween 20 (Sigma, P9416) plus 5% dry milk (Apex, 20–241). Affinity-purified anti-Pat1 antibody was diluted 1:1000 in TBS-T plus 5% dry milk and incubated with the membrane overnight at 4 °C. Anti-RickA[101] was used as a loading control by diluting serum 1:2,000 in TBS-T plus 5% dry milk and incubating at room temperature for 1 h. Membranes were washed with TBS-T for 5 × 5 min intervals at room temperature. Secondary antibody goat anti-rabbit HRP (Santa Cruz Biotechnology, sc-2004) was diluted 1:3,000 in TBS-T plus 5% dry milk and incubated at RT for 30 min, followed by washes with TBS-T. To detect secondary antibodies, ECL HRP substrate kit (Advansta, K-12045) was added to the membrane for 45 s at room temperature and developed using Biomac Light film (Carestream, 178-8207).

**Bacterial infections for imaging**. Infections were carried out in 24-well plates unless otherwise noted. For immunofluorescence microscopy, 24-well plates containing 12 mm sterile coverslips were used. HMECs were seeded at 2.5 × 10$^5$ cells/well and infected 36–48 h later. BMDMs were seeded at 5 × 10$^5$ cells/well and infected 20–24 h later. A549 cells were seeded at 1.2 × 10$^5$ cells/well and infected 24 h later. For time points from 0–2 hpi, an MOI of 3–5 was used for all cell types, and for 24–48 hpi, an MOI of 1-0.05 was used, unless otherwise noted. For the infectious focus assay, an MOI of 0.05 was used. To infect cells, 30% preparations of *R. parkeri* were thawed on ice prior to infection and immediately diluted into fresh media on ice. Cell media was aspirated, the well was washed once with PBS (Gibco, 10010049), 0.5 ml of bacteria in media was added per well, and the plate was spun at 300 × *g* for 5 min at room temperature. Media at 33 °C was added following centrifugation and infected cells were incubated at 33 °C in 5% CO$_2$. For each biological replicate, 2–3 coverslips were infected per strain per condition.

For Gal3 and Gal8 imaging experiments, HMECs were transfected with pmCherry-N1-Gal3 or pmCherry-N1-Gal8 using Lipofectamine LTX (Invitrogen, A12621) and incubated at 37 °C overnight. The next morning, wells were washed twice with PBS and replaced with fresh, warm HMEC media. Cells were visually examined to confirm 80–100% confluency and the presence of Gal3 or Gal8 expression. Infections were performed a few hours later. For quantification, bacteria with Gal staining that was polar or completely overlapping with bacterial staining were considered positive. At least 200 total bacteria in Gal expressing cells were quantified and a minimum of 50 polyUb-, p62-, and NDP52-positive bacteria were quantified to determine the frequency of colocalization with Gal proteins.

The mixed cell assay was adapted from a prior study[69]. Briefly, A549-TRTF cells and unlabeled A549 cells were seeded into 12-well plates at a density of 3 × 10$^5$ cells/ml and grown overnight. The following day, A549-TRTF cells were infected at an MOI of 4 as described above and incubated at 33 °C for 1 h. Both infected A549-TRTF cells and unlabeled A549 cells were detached by adding warm citric saline (135 mM KCl, 15 mM sodium citrate) and incubating for 5 min at 37 °C. Cells were gently resuspended by pipetting up and down and recovered in A549 media, then washed twice with A549 media. Cells were resuspended in A549 media with 10 µg/ml gentamycin to kill extracellular bacteria. Infected A549-TRTF and unlabeled cells were mixed at a ratio of 1:120, plated on coverslips in a 24-well plate, and incubated in a humidified secondary container at 33 °C until 32 hpi.

Hypotonic shock treatment was adapted from a prior study[67]. Briefly, HMECs were infected as stated above and incubated for 5 min at 37 °C, at which point media was exchanged with a hypertonic solution (10% PEG-1000, 0.5 M sucrose in PBS) and incubated at 37 °C for 10 min. Wells were washed gently once with the hypotonic solution (60% PBS), incubated in hypotonic solution at 37 °C for 3 min, then incubated in isotonic HMEC media for 15 min at 37 °C.

**Immunofluorescence microscopy**. Unless otherwise noted, all coverslips were fixed for 10 min in fresh 4% paraformaldehyde (Ted Pella, 18505) at room temperature. Coverslips were washed 3x with PBS pH 7.4 and stored at 4 °C until staining. All antibodies were diluted in PBS with 2% BSA (Sigma, A9418) and all wash steps were done 3x with PBS pH 7.4 after each incubation. All incubations were done at room temperature and all coverslips were mounted in Prolong Gold antifade (Invitrogen, P36930) and sealed with nail polish after drying.

Primary antibodies used to stain *Rickettsia* were rabbit anti-*Rickettsia* I7205[102] (1:300; gift from T. Hackstadt), rabbit anti-*Rickettsia* OmpB[81] (1:1000), and mouse anti-*Rickettsia* 14–13[102] (1:400; gift from T. Hackstadt). Primary antibodies were incubated with coverslips for 30 min. Coverslips were washed and the following secondary antibodies were added for 30 min, protected from light: goat anti-rabbit Alexa 488 (1:400; Invitrogen, A11008), goat anti-rabbit Alexa 404 (1:150; Invitrogen, A31556), goat anti-mouse Alexa 488 (1:400; Invitrogen, A11001), and goat anti-mouse Alexa 404 (1:150; Invitrogen, A31553).

To quantify colocalization with polyubiquitin and autophagy adapters, cells were permeabilized with 0.5% Triton-X100 and washed. Primary antibodies were added for 30 min to 1 h at the following dilutions: mouse anti-polyubiquitin FK1 (1:250; EMD Millipore, 04–262), guinea pig anti-p62 (1:500; Fitzgerald, 20R-PP001), mouse anti-NDP52 (1:300; Novus Biologicals, H00010241-B01P). For staining with rabbit polyclonal anti-LC3 (1:250; Novus Biologicals, NB100-2220SS), mouse anti-human LAMP-1 (1:25; BD Bioscience, 555801), rat anti-mouse LAMP-1 (1:250; BioLegend, 121609), cells were post-fixed in 100% methanol at room temperature for 5 min. After antibody incubations, coverslips were washed and the following secondary antibodies were added for 30 min and protected from the light: goat anti-mouse Alexa 568 (1:500; Invitrogen, A11004), goat anti-mouse Alexa 488 (1:400; Invitrogen, A11001), goat anti-guinea pig Alexa 568 (1:500; Invitrogen, A11075), goat anti-guinea pig Alexa 488 (1:400; Invitrogen, A11073), and goat anti-rat Alexa 568 (1:500; Invitrogen, A11077).

To quantify the percentage of bacteria with actin tails, cells were permeabilized with 0.5% Triton-X100 for 5 min and then washed. *Rickettsia* were stained with either anti-*Rickettsia* 14–13 or anti-*Rickettsia* I7205 as described above followed by a PBS wash. After staining for *Rickettsia*, actin was stained with phalloidin-568 (diluted 1:500 in PBS with 2% BSA; Life Technologies, A12380) for 30 min at room temperature.

To quantify the percentage of bacteria with actin tails in the mixed cell assay, cells were permeabilized with 0.1% Triton-X100 for 5 min and then washed. *Rickettsia* was detected with the primary antibody mouse anti-*Rickettsia* 14–13 and the secondary antibody goat anti-mouse Alexa 404 as described above. After staining for *Rickettsia*, actin was stained with phalloidin-488 (diluted 1:400 in PBS with 2% BSA; Life Technologies, P3457) for 30 min at room temperature and then washed. Coverslips were then washed three times with PBS.

To quantify the size of infectious foci[66,69], cells were permeabilized with 0.05% Triton-X100 for 5 min, washed, and blocked with PBS containing 2% BSA for 1 h. Coverslips were incubated with anti-β-catenin (1:200; BD Bioscience, 610153) for 1 h at room temperature, then washed, followed by incubation with goat anti-mouse Alexa-568 (1:500; Invitrogen, A11004) for 30 min protected from the light. Cells were washed and then subsequently stained to detect *Rickettsia* with anti-I7205 for 30 min and goat anti-rabbit Alexa 488 as described above. Nuclei were stained with Hoechst (1:10,000; Thermo Scientific, 62249) for 15 min. To quantify, the number of infected cells per focus was counted for 10–15 foci per experiment.

To quantify cytosolic and vacuolar bacteria using digitonin permeabilization, we adapted methods from prior studies[14,63,64]. Briefly, HMECs or BMDMs were infected as described above except using an MOI of 5. At 45 mpi, 10 μg/ml gentamycin was added to kill extracellular bacteria. Prior to permeabilization, cells were washed 3× with warm KHM buffer (110 mM potassium acetate, 20 mM HEPES, pH 7.3, 2 mM MgCl₂). HMEC plasma membranes were selectively permeabilized with 70 μg/ml digitonin (Calbiochem, 300410) in KHM for 1 min at room temperature. This digitonin concentration was determined by incubating uninfected HMECs with varying concentrations of digitonin for 1 min at RT and staining for a cytosolic marker, Calnexin (1:200 dilution; Enzo Life Sciences, ADI-SPA860-D), and the ER luminal protein, protein disulfide isomerase (PDI, 1:200 dilution; Enzo Life Sciences, ADI-SPA-890-D). This allowed us to determine the concentration of digitonin that permeabilized the plasma membrane but not intracellular membranes with minimal cell loss/detachment. BMDM plasma membranes were permeabilized with 50 μg/ml digitonin in KHM for 1 min at room temperature (conditions reported previously[63,64]). Following digitonin permeabilization, coverslips were washed gently 3x with KHM buffer, and mouse anti-*Rickettsia* 14–13 was added to coverslips for 15 min at 37 °C to bind cytosolic bacteria. Primary antibody was removed, coverslips were washed 2x with PBS, and goat anti-mouse Alexa 568 was added to coverslips for 15 min at 37 °C to stain cytosolic bacteria. Coverslips were washed 2x with PBS and fixed in 4% paraformaldehyde for 10 min at room temperature. Paraformaldehyde was quenched with 0.1 M glycine for 10 min at room temperature. All bacteria were then stained by washing the cells 3× with PBS with 0.1% saponin and 3% BSA and incubating coverslips with mouse anti-*Rickettsia* 14–13 diluted in PBS with 0.1% saponin and 3% BSA for 1 h at room temperature. Coverslips were washed with PBS, followed by goat anti-mouse Alexa 488 for 30 min at room temperature. Nuclei were stained with Hoechst (1:10,000; Thermo Scientific, 62249) for 15 min and coverslips were then mounted. The number of vacuolar bacteria was determined by subtracting all bacteria from the number of cytosolic bacteria.

For imaging, images were captured on a Nikon Ti Eclipse microscope with a Yokogawa CSU-XI spinning disc confocal, with 60× or 100× (1.4 NA) Plan Apo objectives and a Clara Interline CCD Camera (Andor Technology) using MetaMorph software (Molecular Devices). All experiments captured images as z-stacks and image processing, including for representative images, used z-stack average maximum intensity projections. Unless otherwise note, imaging experiments quantified at least 200 bacteria from three independent replicates. Images were processed in ImageJ and assembled in Adobe Illustrator

**Transmission electron microscopy.** HMEC-1 cells were seeded into six-well plates (1 × 10⁶ cells per well) and grown for 36 h. Media was aspirated and 2.5 ml of bacteria in media at an MOI of 5 for 1 hpi and 0.1 for 48 hpi were added. The plates were spun at 300 × g for 5 min at room temperature, then 2.5 ml of warm HMEC-1 media was added to each well, and the plates were placed at 33 °C. Time points were taken by aspirating media, washing the well with PBS, and fixing the cells in fixative (2% paraformaldehyde, 2% glutaraldehyde in 0.05 M cacodylate buffer, pH 7.2) for 45 min at room temperature. Cells were scraped and pelleted in microcentrifuge tubes and stored in fresh fixative at 4 °C until embedding. Samples were embedded in 2% low melt agarose and placed in 2% glutaraldehyde in 1 M cacodylate buffer, and stored at 4 °C overnight. The next day, samples were post-fixed with 1% osmium tetraoxide and 1.6% potassium ferricyanide, then dehydrated in increasing concentrations of ice-cold ethanol (70–100%). Samples were embedded in Epon 812 resin (11.75 g Epon 12, 6.25 g dodecenyl succinic anhydride, 7 g nadic methyl anhydride, and 0.375 ml of the accelerator benzyl dimethylamine was added during the dehydration step) and stained with 2% uranyl acetate and lead citrate. Images were captured with a FEI Tecani 12 transmission electron microscope and analyzed manually to determine the total number of intracellular bacteria and their respective localization within the cell.

**Mouse studies.** Animal research was conducted under a protocol approved by the University of California, Berkeley Institutional Animal Care and Use Committee (IACUC) in compliance with the Animal Welfare Act and other federal statutes relating to animals and experiments using animals (Welch lab animal use protocol AUP-2016-02-8426-1). Mice were 8–20 weeks old at the time of initial infection. Housing conditions were as follows: lights were turned on from 6 AM-8 PM, temperature was maintained between 20 and 26 °C with a relative humidity of 30–70%. Mice were selected for experiments based on their availability, regardless of sex, and both sexes were used for each experimental group. All mice were of the C57BL/6J background were double knock outs for the genes encoding the receptors for IFN-I (*Ifnar1*) and IFN-γ (*Ifngr1*) (*Ifnar1*⁻/⁻;*Ifngr1*⁻/⁻) (Jackson Labs stock #:029098, described in ref. [61]) and were healthy at the time of infection. *R. parkeri* was prepared by diluting 30% preparation bacteria into 1 ml cold sterile PBS on ice, centrifuging the bacteria at 12,000 × g for 1 min, and resuspending in cold sterile PBS to the desired concentration (5 × 10⁶ PFU/ml for intravenous infection). The bacterial suspensions were kept on ice during injections. Mice were exposed to a heat lamp while in their cages for ~5 min and then each mouse was moved to a mouse restrainer (Braintree, TB-150 STD). The tail was sterilized with 70% ethanol, and 200 μl bacterial suspensions were injected using 30.5-gauge needles into the lateral tail vein. Body temperatures were monitored using a rodent rectal thermometer (BrainTree Scientific, RET-3). Mice were euthanized if their body temperature fell below 90°F (32.2 °C) or if they exhibited severe lethargy that prevented their normal movement around the cage.

**Statistics.** The statistical parameters and significance are reported in the figure legends. Data were considered to be statistically significant when $P < 0.05$, as determined by an unpaired Student's $t$-test (two-tailed), paired Student's $t$-test (two-tailed), a one-way ANOVA with either multiple comparisons or comparison to WT bacteria, a two-way ANOVA, or a log-rank (Mantel-Cox) test. Asterisks denote statistical significance as: *$P < 0.05$; **$P < 0.01$; ***$P < 0.001$; ****$P < 0.0001$, compared with the indicated controls. Statistical analyses were performed using GraphPad PRISM v.9.

**Reporting summary.** Further information on research design is available in the Nature Research Reporting Summary linked to this article.

## Data availability

Source data are provided with this paper. The sequencing reads from whole-genome sequencing of the *R. parkeri* pat1::Tn mutant can be accessed at the NCBI Sequence Read Archive (SRA) as accession number SRR18465981. Source data are provided with this paper.

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

## Acknowledgements

We thank Ted Hackstadt, David Wood, and Christopher Paddock for kindly providing strains and reagents. We thank previous Welch Lab members whose work supported this project, including Rebecca Lamason, Natasha Kafai, Julie Choe, and Shawna Reed, and current lab members for critical feedback throughout the development of this project. We also thank the following UC Berkeley core facilities and their facility members for providing equipment, reagents, and technical support to complete this work: Danielle Jorgens, Reena Zalpuri, and Guangwei Min (UC Berkeley Electron Microscope Laboratory); Holly Aaron and Feather Ives (CRL Molecular Imaging Center); and Alison Killilea (Cell Culture Facility). We thank David Drubin, Karsten Gronert, and Daniel Portnoy for the technical discussion and critical guidance for this work. We also thank Neil Fischer for proofreading the manuscript. This work was funded by grant R01 AI109044 from the NIH/NIAID to M.D.W.

## Author contributions

G.M.B. and M.D.W. designed the study. G.M.B. performed the experiments unless otherwise noted. T.P.B., C.J.T., and P.E. performed the mouse experiments. P.E. aided experimental design for colocalization experiments with polyubiquitin and autophagy markers. N.T.N.L. performed the Sca2 localization experiments. G.M.B. and M.D.W. wrote the manuscript and all authors contributed to edits on the manuscript.

## Competing interests

The authors declare no competing interests.
