## [Peer Review File · Nature Communications]

Reviewers' Comments:

Reviewer #1:

Remarks to the Author:

This is the most in-depth analysis of the role of phospholipases in vacuole escape in Rickettsiaceae performed to date and as such is an important advance for the field. The study is comprehensive and thorough. I have the following specific comments:

1. I remain somewhat confused about why the authors did not observe any growth defect in the pat1 mutant (Fig 1E) despite showing a defect in cell to cell spread. Although the likelihood of redundancy is discussed, it remains unclear why a clear defect in vacuole escape, plaque size and cell spread, as shown in the paper, does not translate to any difference in growth phenotype. Is this true in other cell lines too? Based on the data here the growth curves were carried out up to 96 hours but cell to cell spread was happening by 36 hours. So shouldn't a decrease in cell to cell spread lead to a decrease in bacterial population number?
2. Does parkeri encode Pat2 or only Pat1? Also, is there evidence that PLC does not play a role in Parkeri (rather than other Rickettsia?). For example were there no hits in their previous screen?
3. Information about MOI should be added to the figure legend to make it easier to follow
4. Was an MOI of 0.001 really used for infectious focus assay? This seems very low.
5. Fig 2E – after hypotonic shock the levels of Pat1 mutant bacteria with actin is actually higher than that in wild type. Do the authors have an explanation for this? Was this observed reproducibly?
6. Fig 2C – was this quantification from TEM or IFM? Please state in legend
7. Line 213 – this experiment doesn't technically prove that this effect is caused by secreted pat1. It could also be due to contribution from another rickettsial protein or factor that is acting in a dose-dependent way. The only way to show this was specifically pat1 would be to express this protein in mammalian cells.
8. Fig 7 – I believe legend for C and D are swapped
9. Also was this analysis from TEM or IFM. Please state in legend
10. Line 346 – typo –“by” not “but”
11. Line 628 – what does 30% purified bacteria mean? Please give more information. Was this sample taken from fresh bacteria or frozen stock, how long after infection?
12. Why were A549 cells used for the cell mixing experiments? Please explain.
13. A general comment: It would be helpful to have an overview figure or description of the pat1 gene. Which part is the phospholipase domain? How conserved is it across the Rickettsiaceae? Is there a known regulatory domain and is this present in pat1 from Rp?
14. A general question – how does Rp avoid membrane damage caused by overactivation of the pat1 gene? Is there anything known about regulation? Is there any data on expression – for example when it is expressed during infection?
15. Another general question - could you elaborate a little more on how you can measure the effect on cell to cell spread in a mutant that is impaired in the initial vacuole exit? I assume that enough bacteria sneak through to be able to measure this but it would be helpful to have this clarified a little in the manuscript. In Fig 7D the percentage of bacteria with plasma membrane goes from 1% in WT to 12% in mutant but it would be helpful to have the numbers too. I assume it was much harder to find bacteria to count in the mutant case, because the overall number was lower. It would be helpful to know the number counted.
16. Throughout, the number of bacteria counted when a percentage is calculated is not always given. This information is required.
17. non-parametric Mann Whitney test should be used to determine significant differences between small sample numbers.

Reviewer #2:

Remarks to the Author:

The manuscript reports that patatin-like phospholipase A2 enzyme (Pat1) plays critical roles in escaping from host membranes and promoting cell-cell spread during *R. parkeri* infection. To demonstrate the role of Pat1, the authors used a *R. parkeri* pat1::Tn mutant that they constructed in their earlier work. Overall, the manuscript is well written with clear methodology, and a

significant amount of works has been presented to demonstrate the role of Pat1 in *R. parkeri* infection of host. However, the data presented, does not convincingly support the conclusion that *R. parkeri* Pat1 mediates escape from host membranes, evasion of host defense surveillance and cell-cell spread. Some data are contradictory. Most importantly, the manuscript did not provide data showing phospholipase A2 enzymatic activity of Pat1 molecule from *R. parkeri*, a member of spotted fever group *Rickettsia* species. The authors need to address the comments, as detailed below.

Specific Comments:

1. *R. parkeri* is a member of spotted fever group (SFG) *Rickettsia*. This is a first report on Pat1 molecules from SFG *Rickettsia* spp. However, the most important data on phospholipase A2 enzyme activity of *R. parkeri* Pat1 protein, is missing in this manuscript.
2. It is important to demonstrate the Pat1 phospholipid substrate specificity and host factor(s) required for Pat1 enzymatic activity, to implicate its role in escaping from host membranes.
3. Fig.1A. Did the authors check the transposon insertion in *pat1::Tn* mutant in one or multiple location of genome? The authors need to address the effect on adjacent genes by transposon insertion in the genome of *pat1::Tn* and *pat1::Tn pat1+* mutants.
4. Fig.1B. The band near 75 kD should be addressed.
5. Fig 1C. The authors emphasized that the *pat1:Tn* mutant was identified based on its small-plaque phenotype, however the presented plaques image of *pat1:Tn* mutant in Fig 1C is not of sufficient quality and that makes it difficult to compare with plaques of WT and *pat1::Tn pat1+* mutant.
6. Fig.1E. The growth curve using HMEC host should be accompanied with data showing that host cells were well infected with all three *Rickettsia* strains (e.g., western blot showing, Pat1, any house keeping gene of *Rickettsia*, as well as, host cell house keeping gene).
7. Given the importance of both endothelial cells and macrophages to *R. parkeri* infection, the authors should characterize (e.g., growth/replication, etc.) the identified Pat1 mutant in both cell types to strengthen the biological significance of their findings.
8. In vivo contributing role of Pat1: The authors used IFN-I/ IFN- γ double knockout mice for infection studies, however, no data was provided showing the pathology for WT or mutant *Rickettsia* strains during the course of infection. To demonstrate rickettsial in vivo dissemination, no bacterial burden in spleens, livers was provided as well as no histology data. In addition, the authors should also assess the role of Pat1 in IFN-I or IFN- γ individual knockout mice.
9. Fig.2. The TEM images showed in Fig.2A should reflect the data presented in Fig. 2B (e.g., WT vs mutant in the cytosol and not just the counting of differences in double and single membrane appearance). Representative TEM images for Fig.2D and E should be presented to strengthen the authors findings.
10. Fig 3: Representative images of polyUb staining of *pat1::Tn pat1+* mutant in HMEC should be included to help comparing the difference in polyUb+ staining of WT vs *pat1::Tn* mutant bacteria. The authors should also perform the similar staining assay in macrophages to strengthen their findings.
11. Fig.4. Western blot analysis showing the expression profiles of the utilized autophagy markers (p62, NDP52 and LC3) in uninfected and bacteria-infected lysates at different time points should be presented.
12. Fig.S1. Line 202-204. Increased colocalization of *pat1::Tn* mutant with Lamp-1 data would suggest that the *pat1:Tn* mutant is targeted and destroyed by lysosomal pathway, which is supported by in vivo survival data (Fig.1F). However, the data in Fig 1E shows no differences in rickettsial survival/replication up to 96 hrs for WT, *pat1::Tn* and *pat1::Tn pat1+* -complemented strains in HMEC cells. How do the authors reconcile and interpret these contradictory data? Again, experiments using macrophages would provide additional insights into how Pat1 functions as an essential virulent factor during infection.
13. Fig.5. The authors showed that Pat1 function is required for avoiding bacterial association with damaged membranes using HMEC cells. To provide more insights on the role of Pat1, it will be helpful to perform the similar assay using macrophages.
14. Fig.6/7. Again, it will be helpful to provide data using macrophages.

Minor Comment:

1. All presented images (e.g., Fig 1C, Fig 3, Fig 4, Fig 5, etc.) should have their own individual scale bar.

Reviewer #3:

Remarks to the Author:

This manuscript uses genetic tools (transposon mutagenesis) to confirm previously suggested roles for a *Rickettsia parkeri* patatin-like phospholipase A2 (Pat1) in escape from the initial endocytic vacuole into the host cell cytosol where the rickettsiae replicate. The work appears to be carefully done and will be useful information in the field. The data for the role of Pat1 in vacuolar escape are convincing, however, some of the additional effects described for the Pat1 knockout seem like they could be indirect or downstream effects of this disruption. I believe additional information or discussion is needed to bring some needed clarity to the direct function of Pat1. Specific comments are listed below.

1. Fig 1. The WT and Pat1::tn mutant replicate essentially equally. I assume the rickettsiae are not replicating within the vacuole but gain access to the cytosol. It seems likely that the rate of escape from the endocytic vacuole differs but that enough rickettsiae enter the cytosol quickly enough to be able to replicate at the same rate. Quantitation of rickettsiae remaining within vacuoles is done at 1 hr post-infection. It would be useful to have additional information on the kinetics of this escape by examining more time points to see what has happened by 2 or 4 hr post-infection. This is addressed somewhat in the Discussion (lines 358-360) with the suggestion that Pat1 is likely one of multiple redundant mechanisms to escape the vacuole but if data is available, that would be helpful.

2. Fig 2. There are several statements throughout the manuscript, beginning in the Abstract, that Pat1 plays critical roles in promoting cell-cell spread. The Figure and lines 168-171 indicate that actin association with the rickettsia was determined at 30 and 60 min. post-infection. This is quite early and the lack of association with actin seems likely due to the rickettsia being sequestered in a vacuole. Lines 248-262 address later interactions with actin and cell-cell spread, but again, one wonders if this is simply due to a reduced ability to escape from the double membrane after rickettsiae penetrate adjacent cells.

3. Overall, I'm convinced that Pat1 plays a role in escape from the endocytic vesicle but I am less convinced that Pat1 performs several critical functions as stated on lines 46-47. Avoiding polyubiquitination, preventing recruitment of autophagy markers, and promoting actin polymerization don't seem like direct functions of Pat1 and should be put into perspective.

Response to Reviewers

We thank the reviewers for taking the time to thoroughly read and review our manuscript and for making very thoughtful and constructive comments. The comments helped us incorporate key improvements into this revised manuscript.

Reviewer #1:

General comments:

“This is the most in-depth analysis of the role of phospholipases in vacuole escape in Rickettsiaceae performed to date and as such is an important advance for the field. The study is comprehensive and thorough.”

Response: We thank the reviewer for this positive comment and appreciate their recognition of the importance of our findings for the field.

Specific comments:

1. *“I remain somewhat confused about why the authors did not observe any growth defect in the *pat1* mutant (Fig 1E) despite showing a defect in cell to cell spread. Although the likelihood of redundancy is discussed, it remains unclear why a clear defect in vacuole escape, plaque size and cell spread, as shown in the paper, does not translate to any difference in growth phenotype. Is this true in other cell lines too? Based on the data here the growth curves were carried out up to 96 hours but cell to cell spread was happening by 36 hours. So shouldn't a decrease in cell to cell spread lead to a decrease in bacterial population number?”*

Response: To further address this question related to how the *pat1::Tn* mutant is able to grow normally and whether this is true in other cell types, more data has been added to the manuscript to demonstrate that (1) growth is impaired in primary mouse bone marrow-derived macrophages (BMDMs) (Fig. 1E) and (2) the *pat1::Tn* mutant is delayed in accessing the cytosol at 1.5 hpi in HMECs, but at 4 hpi it is able to access the cytosol at a similar frequency as WT and the complemented strain (Fig. 2C, D).

With regard to whether a defect in cell-to-cell spread should affect bacterial growth, our previous work [for example, references 66 and 69] demonstrated that strains with mutations in the *sca2* gene, which fail to undergo actin-based motility and have diminished cell-cell spread, nevertheless grow normally in epithelial cells. Therefore, a defect in cell-cell spread is not necessarily correlated with a growth defect. Similarly, we previously showed that *Rickettsia* mutants in *ompB* that are targeted by the host autophagy machinery grow similarly to WT in HMECs [reference 83].

2. *“Does *parkeri* encode Pat2 or only Pat1? Also, is there evidence that PLC does not play a role in *Parkeri* (rather than other *Rickettsia*?). For example were there no hits in their previous screen?”*

Response: *R. parkeri* only encodes Pat1 and we have clarified this in line 57-59 of the revised manuscript. Additionally, there are no predicted/annotated PLC enzymes encoded in any *Rickettsia* species. As for phospholipase D (PLD), no mutants in the *pld*

gene were isolated our transposon mutagenesis screens in *R. parkeri* [reference 60]. Furthermore, a PLD mutant in *R. prowazekii* did not show a defect in vacuolar escape or plaque size [reference 30].

3. *“Information about MOI should be added to the figure legend to make it easier to follow.”*

Response: As suggested, information about specific MOI used for bacterial infections were added to all figure legends.

4. *“Was an MOI of 0.001 really used for infectious focus assay? This seems very low.”*

Response: Thank you for drawing this to our attention. The 0.001 value was incorrect and the correct MOI is 0.05. We have changed line 739 in the materials and methods section accordingly.

5. *“Fig 2E – after hypotonic shock the levels of Pat1 mutant bacteria with actin is actually higher than that in wild type. Do the authors have an explanation for this? Was this observed reproducibly?”*

Response: This was observed reproducibly. We think the number of actin tails might be higher for the *pat1::Tn* mutant following hypotonic shock because the shock treatment acts to synchronize escape of the mutant bacteria that have accumulated in intact vacuoles. Upon synchronized release we believe they will initiate actin-based motility at higher frequency than observed for wild-type bacteria which are released asynchronously.

6. *“Fig 2C – was this quantification from TEM or IFM? Please state in legend”*

Response: Invasion was quantified by immunofluorescence microscopy. This data was moved to supplementary Fig. S2. We added immunofluorescence microscopy image panels and clarified the method used in the legend and in line 174 in the results section.

7. *“Line 213 – this experiment doesn’t technically prove that this effect is caused by secreted pat1. It could also be due to contribution from another rickettsial protein or factor that is acting in a dose-dependent way. The only way to show this was specifically pat1 would be to express this protein in mammalian cells.”*

Response: We thank the reviewer for this feedback. We have changed the wording of our interpretation of this result in line 234-236 to: “These results indicate that the presence of WT bacteria counteracts targeting of the *pat1::Tn* mutant bacteria by ubiquitylation and recruitment of the autophagy machinery.”

8. *“Fig 7 – I believe legend for C and D are swapped”*

Response: We thank the reviewer for bringing this to our attention. The figure legend was incorrect and we have corrected the mistake.

9. “Also was this analysis from TEM or IFM. Please state in legend”

Response: Quantification of vacuolar and cytosolic bacteria was done from TEM images (representative images in panel A). Panel C and D were quantified using IFM images. We have clarified the methods used in the figure legend and in line 311 and line 323.

10. “Line 346 – typo – ‘by’ not ‘but’”

Response: Thank you for bringing to our attention, we have corrected the typo on line 367 of the revised manuscript

11. “Line 628 – what does 30% purified bacteria mean? Please give more information. Was this sample taken from fresh bacteria or frozen stock, how long after infection?”

Response: The preparation of 30% purified bacteria is described in Materials and Methods under the section “*R. parkeri* strains and bacterial isolation,” line 545-558. We also clarified that infections were done by thawing frozen stocks on line 556-557 and under the section “Bacterial infections for imaging” line 739-740.

12. “Why were A549 cells used for the cell mixing experiments? Please explain.”

Response: A549 cells were used because they are good for imaging and because we had an existing cell line that stably expresses the TagRFP-T-farnesyl plasma-membrane marker. This cell line was used in one of our previous publications to examine *R. parkeri* cell-cell spread [reference 69].

13. “A general comment: It would be helpful to have an overview figure or description of the *pat1* gene. Which part is the phospholipase domain? How conserved is it across the *Rickettsiaceae*? Is there a known regulatory domain and is this present in *pat1* from *Rp*?”

Response: We have added a new Fig. S1A showing an alignment of the amino acid sequence of Pat1 from *R. parkeri* and *R. typhi* along with a representative patatin from *Solanum tuberosum*. Key sequence elements and catalytic residues are also indicated. In response to comments from Reviewer #2, we also show data in Fig. S1B demonstrating Pat1 enzymatic activity *in vitro*. There are currently no known regulatory domains for Pat1.

14. “A general question – how does *Rp* avoid membrane damage caused by overactivation of the *pat1* gene? Is there anything known about regulation? Is there any data on expression – for example when it is expressed during infection?”

Response: This is an excellent question. How *R. parkeri* Pat1 is regulated is unknown, although our new data in Fig. S1B suggest that Pat1 activity is enhanced by a component in host cells. This is similar to prior findings that *R. typhi* Pat1/Pat2 are secreted in host cell cytosol at 24 hpi and that activity *in vitro* requires cell lysates or host superoxide dismutase SOD [references 28 and 37]. Furthermore, a recent publication did find that ubiquitin can activate *Rickettsia prowazekii* Pat2 (Tessmer *et al*, *J Bacteriology*, 2019). Although beyond the scope of this work, future studies will focus on the regulation of phospholipase activity during infection.

15. *“Another general question - could you elaborate a little more on how you can measure the effect on cell to cell spread in a mutant that is impaired in the initial vacuole exit? I assume that enough bacteria sneak through to be able to measure this but it would be helpful to have this clarified a little in the manuscript. In Fig 7D the percentage of bacteria with plasma membrane goes from 1% in WT to 12% in mutant but it would be helpful to have the numbers too. I assume it was much harder to find bacteria to count in the mutant case, because the overall number was lower. It would be helpful to know the number counted.”*

Response: We thank the reviewer for this excellent question. As mentioned previously in response to specific comment 1, we have provided additional data demonstrating that the *pat1::Tn* mutant accesses the cytosol similar to WT and the complemented strain at later time points (4 hpi) in HMECs (Fig. 2D). The data provides evidence to support that the initial defect in early vacuolar escape resolves by 4 hpi, suggesting early vacuolar escape is not directly impacting later events such as replication and spread (discussed in line 170-172 and line 391-392).

In our experience, the relative bacterial number at later timepoints is similar for WT and the mutant. Indeed, the difference we see is what you noted where there are more mutant bacteria in the primary cell compared to neighboring cells. In other words, the mutant bacteria grow to higher density in the primary cell. We have added the number of bacteria found in the primary vs secondary cell for each strain to Fig. 6E.

16. *“Throughout, the number of bacteria counted when a percentage is calculated is not always given. This information is required.”*

Response: We have added the number of bacteria counted to each figure legends.

17. *“Non-parametric Mann Whitney test should be used to determine significant differences between small sample numbers.”*

Response: We thank the reviewer for bringing our attention to this. We have performed a non-parametric Mann Whitney test for the experiments quantifying actin tails and localization in secondary cells, since the number of events counted for the mutant was less than 30. Please let us know if you were referring to a different data set and we will correct the error.

Reviewer #2:**General comments:**

“The manuscript reports that patatin-like phospholipase A2 enzyme (Pat1) plays critical roles in escaping from host membranes and promoting cell-cell spread during R. parkeri infection. To demonstrate the role of Pat1, the authors used a R. parkeri pat1::Tn mutant that they constructed in their earlier work. Overall, the manuscript is well written with clear methodology, and a significant amount of works has been presented to demonstrate the role of Pat1 in R. parkeri infection of host. However, the data presented, does not convincingly support the conclusion that R. parkeri Pat1 mediates escape from host membranes, evasion of host defense surveillance and cell-cell spread. Some data are contradictory. Most importantly, the manuscript did not provide data showing phospholipase A2 enzymatic activity of Pat1 molecule from R. parkeri, a member of spotted fever group Rickettsia species. The authors need to address the comments, as detailed below.”

Response: We thank the reviewer for their critical reading of our manuscript. In response to the reviewer’s suggestion that it was most important for us “provide data showing phospholipase A2 enzymatic activity of Pat1 molecule from *R. parkeri*,” we performed an *in vitro* PLA₂ assay similar to what was previously used for *R. typhi* Pat1 and Pat2 (Fig. S1B) [references 28, 37]. Importantly, we found that purified recombinant *R. parkeri* Pat1 alone has PLA₂ activity, whereas purified recombinant Pat1 with a mutation in the catalytic serine at position 50 (Pat1 S50A) does not. Furthermore, this activity is enhanced by the addition of host cell lysate, similar to what was observed for *R. typhi* Pat1 and Pat2. Thus, we can now definitively say that *R. parkeri* Pat1 is a PLA₂ enzyme.

Specific comments:

1. *“R. parkeri is a member of spotted fever group (SFG) Rickettsia. This is a first report on Pat1 molecules from SFG Rickettsia spp. However, the most important data on phospholipase A2 enzyme activity of R. parkeri Pat1 protein, is missing in this manuscript.”*

Response: We thank the reviewer for pointing out the novelty of examining Pat1 function during SFG infection. As mentioned above, we now provide data in Fig. S1B that definitively shows that *R. parkeri* Pat1 is a PLA₂ enzyme.

2. *“It is important to demonstrate the Pat1 phospholipid substrate specificity and host factor(s) required for Pat1 enzymatic activity, to implicate its role in escaping from host membranes.”*

Response: We agree that it will be interesting to determine the host phospholipid substrates of Pat1 as well as the host proteins that enhance its activity. However, we respectfully argue that this would constitute a separate study that is beyond the scope of our current study. Given that these experiments would take a considerable amount of work and do not add to our key conclusions, we feel this should be addressed in future work.

3. *“Fig.1A. Did the authors check the transposon insertion in pat1::Tn mutant in one or multiple location of genome? The authors need to address the effect on adjacent genes by transposon insertion in the genome of pat1::Tn and pat1::Tn pat1+ mutants.”*

Response: We have sequenced the whole genome of the *pat1::Tn* mutant and there is a single transposon insertion in *pat1::Tn* (raw sequencing reads from WGS are uploaded onto NCBI’s Sequence Read Archive (SRA) server (SRA run accession: SRR18465981) and this information is provided in the section “data availability” in line 910-913. With regard to how the transposon insertion in *pat1* affects adjacent genes, the fact that the *pat1::Tn pat1+* complementation rescues all phenotypes investigated demonstrates that the phenotypes observed for *pat1::Tn* are due to disruption of *pat1*. With regard to the possible effects of the second transposon insertion in the *pat1::Tn pat1+* strain, our previous work [reference 60] identified a transposon insertion in the gene downstream of the *pat1+* insertion that resulted in a reduced plaque size. Therefore, if the second transposon insertion was having a polar effect, we would expect to see a small plaque in the *pat1::Tn pat1+* strain. Instead, we report a plaque size similar to WT like we observe (Fig. 1D). Thus, we conclude that the second transposon insertion is not having an effect on adjacent genes.

4. *“Fig.1B.The band near 75 kD should be addressed.”*

Response: We now note the non-specific 75 kD band in the figure legend.

5. *“Fig 1C. The authors emphasized that the pat1:Tn mutant was identified based on its small-plaque phenotype, however the presented plaques image of pat1:Tn mutant in Fig 1C is not of sufficient quality and that makes it difficult to compare with plaques of WT and pat1::Tn pat1+ mutant.”*

Response: To address this concern we added improved plaque images for the *pat1::Tn* mutant in Fig. 1C.

6. *“Fig.1E. The growth curve using HMEC host should be accompanied with data showing that host cells were well infected with all three Rickettsia strains (e.g., western blot showing, Pat1, any house keeping gene of Rickettsia, as well as, host cell house keeping gene).”*

Response: We appreciate the suggestion. Because our growth curves were conducted by measuring plaque forming units, a control showing host cells are infected is not necessary, as we would be unable to isolate live bacteria if they did not infect the host cells. Thus, our data already demonstrates that the cells are infected with each strain and a western blot would not further strengthen the conclusions.

7. *“Given the importance of both endothelial cells and macrophages to R. parkeri infection, the authors should characterize (e.g., growth/replication, etc.) the identified Pat1 mutant in both cell types to strengthen the biological significance of their findings.”*

Response: We agree with the reviewer that the function of Pat1 is better understood by examining its role in multiple cell types. To address this concern, we performed the growth curve (Fig. 1F), LAMP-1 staining (Fig. S3C, D), and an alternative vacuolar escape assay in macrophages (Fig. 2E) to address the function of Pat1 in different cell types.

8. *“In vivo contributing role of Pat1: The authors used IFN-I/ IFN- γ double knockout mice for infection studies, however, no data was provided showing the pathology for WT or mutant Rickettsia strains during the course of infection. To demonstrate rickettsial in vivo dissemination, no bacterial burden in spleens, livers was provided as well as no histology data. In addition, the authors should also assess the role of Pat1 in individual knockout mice.”*

Response: This study focuses on the cellular level functions of Pat1. We included the animal data (Fig. 1G, H, I) to make a key point that Pat1 is important for infection in vivo. We agree with the reviewer that such follow-up studies will be interesting and important. However, we respectfully assert that a detailed characterization of bacterial burdens in different organs as well as histology data is beyond the scope of our study and is best addressed in future work. Furthermore, with regard to the suggestion to infect single IFN-I or IFN- γ knockout mice, these studies would be of limited utility because our previous work demonstrated that IFN-I or IFN- γ single knockout mice do not succumb to infection with WT *R. parkeri* [reference 61].

9. *“Fig.2. The TEM images showed in Fig.2A should reflect the data presented in Fig. 2B (e.g., WT vs mutant in the cytosol and not just the counting of differences in double and single membrane appearance). Representative TEM images for Fig.2D and E should be presented to strengthen the authors findings.”*

Response: This comment was similar to Reviewer #1's comment 6 regarding how panels 2C and 2D were quantified. Fig. 2A does include representative examples of both cytosolic and vacuolar bacteria for both WT and the *pat1::Tn* mutant. Fig. 2C (now Fig. S2) was quantified by immunofluorescence microscopy and now has image panels added and the method is clarified in the legend and in line 174. Fig. 2D (now Fig. 2G) was also quantified by immunofluorescence microscopy, not by TEM. We have clarified this by adding representative immunofluorescence microscopy images (Fig. 2F), specifying in the figure legend and in line 186.

10. *“Representative images of polyUb staining of *pat1::Tn pat1+* mutant in HMEC should be included to help comparing the difference in polyUb+ staining of WT vs *pat1::Tn* mutant bacteria. The authors should also perform the similar staining assay in macrophages to strengthen their findings.”*

Response: Representative images for polyUb staining of *pat1::Tn pat1+* have been added to Fig. 3A. We have also added data of polyUb+ bacteria in macrophages at 1 hpi for WT and *pat1::Tn* infected cells (Fig. 3D)

11. “Western blot analysis showing the expression profiles of the utilized autophagy markers (p62, NDP52 and LC3) in uninfected and bacteria-infected lysates at different time points should be presented.”

Response: We appreciate the reviewer’s comment. However, our intention is to look at autophagic targeting of the bacteria, which is done using immunofluorescence microscopy as we show in Fig. 4. It is not our intention to measure the effect of infection on the autophagy pathway, which could be done by western blotting. Such measurements would constitute a separate study and are beyond the scope of our manuscript.

12. “Fig.S1. Line 202-204. Increased colocalization of *pat1::Tn* mutant with Lamp-1 data would suggest that the *pat1::Tn* mutant is targeted and destroyed by lysosomal pathway, which is supported by in vivo survival data (Fig.1F). However, the data in Fig 1E shows no differences in rickettsial survival/replication up to 96 hrs for WT, *pat1::Tn* and *pat1::Tn pat1+* -complemented strains in HMEC cells. How do the authors reconcile and interpret these contradictory data? Again, experiments using macrophages would provide additional insights into how Pat1 functions as an essential virulent factor during infection.”

Response: Our observation that the *pat1* mutant grows similarly to WT bacteria in HMEC cells but nevertheless has a growth defect in vivo is not contradictory but in fact is similar to our other published observations for the *R. parkeri ompB* (outer membrane protein B) mutant [reference 83] as well as other literature demonstrating that HMECs aren’t able to eliminate bacteria as efficiently as other cell types [reference 84]. In response to Reviewer #1 comment 1, we also provided additional data showing that the *pat1::Tn* mutant accesses the cytosol with similar levels as WT and complement strains by 4 hpi (prior to bacterial replication)(Fig. 2D), providing an explanation for why the observed delay in vacuolar escape does not impact growth in HMECs. As suggested, we also repeated the growth curve in primary murine BMDMs and found that *pat1::Tn* does not replicate as robustly as WT or the complement strains (Fig. 1F). Furthermore, as suggested in this comment, we also provided new LAMP-1 colocalization data in macrophages (Fig. S3C, D), demonstrating that *pat1::Tn* is targeted to LAMP-1-positive compartments in this cell type.

13. “Fig.5. The authors showed that Pat1 function is required for avoiding bacterial association with damaged membranes using HMEC cells. To provide more insights on the role of Pat1, it will be helpful to perform the similar assay using macrophages.”

Response: To address this suggestion we performed a digitonin permeabilization assay in both HMECs and macrophages (Fig. 2D, E). Digitonin selectively permeabilizes the host cell plasma membrane but not internal membranes, allowing us to determine the percent of vacuolar and cytosolic bacteria following invasion. We observed more vacuolar *pat1::Tn* mutant bacteria compared to WT at 1.5 hpi, confirming that Pat1 contributes to vacuolar escape following entry into BMDMs. Because we use primary BMDMs in our experiments, which are not amenable to

transient transfection, we could not perform galectin colocalization experiments. We agree that further characterizing the dynamics of galectin recruitment in different cell types will be interesting but this will need to be conducted in future studies.

14. *“Fig.6/7. Again, it will be helpful to provide data using macrophages.”*

Response: We appreciate the reviewer’s suggestion. However, we cannot find any published examples of studies of cell-cell spread in macrophage monoculture. Furthermore, it is unclear what additional information or biological insight would be gained by studying cell-cell spread and actin-based motility in this cell type beyond what we have shown in other cultured cells. For this reason, we feel such experiments would not further contribute to the conclusions of our study.

Minor comment:

“All presented images (e.g., Fig 1C, Fig 3, Fig 4, Fig 5, etc.) should have their own individual scale bar.”

Response: Scale bars have been added to all images.

Reviewer #3:

General comments:

*“This manuscript uses genetic tools (transposon mutagenesis) to confirm previously suggested roles for a *Rickettsia parkeri* patatin-like phospholipase A2 (Pat1) in escape from the initial endocytic vacuole into the host cell cytosol where the rickettsiae replicate. The work appears to be carefully done and will be useful information in the field. The data for the role of Pat1 in vacuolar escape are convincing, however, some of the additional effects described for the Pat1 knockout seem like they could be indirect or downstream effects of this disruption. I believe additional information or discussion is needed to bring some needed clarity to the direct function of Pat1.”*

Response: We thank the reviewer for their positive comments about our work. We have provided additional information to support a direct role for Pat1 in the kinetics of vacuolar escape, as detailed below. Furthermore, we clarify in our discussion that we conclusively demonstrate a primary role for Pat1 in escape from the vacuole, which can lead to secondary consequences on other pathways such as autophagy targeting and actin-based motility. Moreover, we can detect other effects of Pat1 that do not seem to be explained by a simple failure to efficiently escape the vacuole. However, as the reviewer suggests, we cannot conclude that these are due to a primary role for Pat1 in these processes. We have therefore worked to clarify what we can conclude are primary versus secondary functions of Pat1 throughout the manuscript. To address this general comment the following phrases were added to the Discussion section:

Line 344-346 *“Altogether, these data suggest Pat1 plays a primary role in vacuolar escape and is of importance at multiple steps of the *Rickettsia* life cycle that involve manipulating host membranes”*

Line 413-414 “Nevertheless, our results suggest that Pat1 enables efficient vacuolar escape which allows for reduced detection and targeting by components of the autophagy pathway.”

Line 418-420 “To what extent a role for Pat1 in avoidance of autophagic targeting of bacteria independent of membrane damage is a secondary consequence Pat1 function in vacuolar escape or a more direct function of Pat1 remains unclear”

Line 433-436 “The *pat1::Tn* mutant also formed fewer actin tails in secondary infected cells following cell-cell spread, suggesting at least some of the reduction in actin tails at late time points is due to reduced escape from the secondary vacuole.”

Line 449-451 “Our data demonstrate that Pat1 plays important roles throughout the *R. parkeri* intracellular life cycle, with a principal role in escaping from vacuoles, and additional contributions to avoiding autophagy and enabling cell-cell spread. Whether Pat1 directly performs other critical functions during infection remains to be determined.”

Specific comments:

1. “Fig 1. The WT and *Pat1::tn* mutant replicate essentially equally. I assume the rickettsiae are not replicating within the vacuole but gain access to the cytosol. It seems likely that the rate of escape from the endocytic vacuole differs but that enough rickettsiae enter the cytosol quickly enough to be able to replicate at the same rate. Quantitation of rickettsiae remaining within vacuoles is done at 1 hr post-infection. It would be useful to have additional information on the kinetics of this escape by examining more time points to see what has happened by 2 or 4 hr post-infection. This is addressed somewhat in the Discussion (lines 358-360) with the suggestion that Pat1 is likely one of multiple redundant mechanisms to escape the vacuole but if data is available, that would be helpful.”

Response: We thank the reviewer for their thoughtful observation and suggestion. To address this comment, as well as Reviewer #3’s comment 13, we performed a digitonin permeabilization assay, which selectively permeabilizes the host cell plasma membrane but not internal membranes. This allowed us to quantify the percentage of bacteria that remained in the vacuole or escaped into the cytosol at 1.5 h and 4 h post of infection (Fig. 2D). We observed that a higher percentage of *pat1::Tn* bacteria were in vacuoles at the early time point compared to WT or *pat1::Tn pat1+* complemented bacteria, but the percentages of bacteria in vacuoles/cytosol were not significantly different by 4 hpi. This suggests that *pat1* mutant bacteria are slower to escape from the vacuole into the cytosol, but that enough escape at later timepoints to grow normally in HMECs.

2. “Fig 2. There are several statements throughout the manuscript, beginning in the Abstract, that Pat1 plays critical roles in promoting cell-cell spread. The Figure and lines 168-171 indicate that actin association with the rickettsia was determined at 30 and 60 min. post-infection. This is quite early and the lack of association with actin seems likely due to the rickettsia being sequestered in a vacuole. Lines 248-262 address later interactions with actin and cell-cell spread, but again, one wonders if this is simply due

to a reduced ability to escape from the double membrane after rickettsiae penetrate adjacent cells.”

Response: We agree with the interpretation that the early reduction in actin tails is a secondary consequence of the *pat1::Tn* mutant being trapped in the vacuole and sequestered away from actin in the host cell cytosol, as stated in line 198-199. Based on our data looking at cell-cell spread later in infection, we demonstrate that the *pat1::Tn* mutant forms fewer actin tails in the primary cell (Fig. S5C) and does not spread into neighboring cells as frequently (Fig. 6E), suggesting that even at later time points at least some of the reduction in actin tails is occurring in the primary cell. This is further supported by our new data demonstrating that by 4 hpi, equal numbers of WT and the *pat1::Tn* mutant bacteria are in the cytosol (Fig. 2D). We also do see a difference in the percentage of actin tails formed in the secondary cell (Fig. S5C), suggesting that there is some contribution of impaired escape from double membrane vacuoles, which we have noted in the discussion section line 433-436. Future work will need to be done to fully clarify whether the reduced frequency of actin tails can be fully accounted for by a failure to escape from primary or secondary vacuoles, or whether there are additional functions of Pat1 that contribute to actin-based motility.

3. *“Overall, I’m convinced that Pat1 plays a role in escape from the endocytic vesicle but I am less convinced that Pat1 performs several critical functions as stated on lines 46-47. Avoiding polyubiquitination, preventing recruitment of autophagy markers, and promoting actin polymerization don’t seem like direct functions of Pat1 and should be put into perspective.”*

Response: We agree with the reviewer that more discussion/perspective should be given about how vacuolar escape relates to the other phenotypes observed. As noted above in our response to this reviewer’s general comments, we have added clarifications to the Discussion section to more clearly spell out what we can conclude are primary versus secondary functions of Pat1. Line 46-47 referred to above has been removed (Importance section not relevant for manuscript formatting) and the abstract has also been modified with previous line 34 (“...and suggest diverse roles for patatin-like phospholipases in facilitating microbial infections.” being deleted to clarify we aren’t concluding from our data that Pat1 has multiple functions.

Reviewers' Comments:

Reviewer #1:

Remarks to the Author:

The authors have addressed all my comments. Really nice study!

Reviewer #2:

Remarks to the Author:

The manuscript is substantially improved from the original submission with better clarity and additional experimental data including PLA2 enzymatic activity. The authors have nicely addressed my comments.

Reviewer #3:

Remarks to the Author:

I am satisfied with the modifications and feel like these have really improved the manuscript. The only suggestion I would make would be to consider moving Supplementary figure 1 into the main body of the paper. Actual demonstration of enzymatic activity is an important point.

Response to Reviewers NCOMMS-21-43151B

Reviewer #1 (Remarks to the Author):

“The authors have addressed all my comments. Really nice study!”

Response: We thank the reviewer for their positive comment.

Reviewer #2 (Remarks to the Author):

“The manuscript is substantially improved from the original submission with better clarity and additional experimental data including PLA2 enzymatic activity. The authors have nicely addressed my comments.”

Response: We thank the reviewer for their positive comments.

Reviewer #3 (Remarks to the Author):

“I am satisfied with the modifications and feel like these have really improved the manuscript. The only suggestion I would make would be to consider moving Supplementary figure 1 into the main body of the paper. Actual demonstration of enzymatic activity is an important point.”

Response: As suggested by the reviewer, we have now moved the enzymatic activity data from the supplement to the main Figure 1 as Fig. 1a.